# COMPUTE-CONSTRAINED DATA SELECTION

**Junjie Oscar Yin**
Johns Hopkins University
jyin27@jhu.edu

**Alexander M. Rush**
Cornell University
arush@cornell.edu

## ABSTRACT

Data selection can reduce the amount of training data needed to finetune LLMs; however, the efficacy of data selection scales directly with its compute. Motivated by the practical challenge of compute-constrained finetuning, we consider the setting in which both the cost of selecting data and training are budgeted for. We first formalize the problem of data selection with a cost-aware utility function, and model the data selection problem as trading off initial-selection cost for training gain. We run a comprehensive sweep of experiments across multiple tasks, varying compute budget by scaling finetuning tokens, model sizes, and data selection compute. Interestingly we find that many powerful data selection methods are almost never compute-optimal, and that cheaper data selection alternatives dominate both from a theoretical and empirical perspective. For compute-optimal training, we find that perplexity and gradient data selection require training-to-selection model size ratios of 5x and 10x, respectively.

## 1 INTRODUCTION

The growth of large language models (LLMs) has motivated research into their resource profiles. The compute cost of training LLMs is substantial, and, in many cases, the total compute budget is predetermined: the number of accelerators and their usage hours are allocated in advanced. Thus, it is critical to determine the optimal allocation of resources under a budget constraint. Past work on compute-optimal LLMs (Hoffmann et al., 2022) studies if one could attain a better perplexity for a given pretraining compute budget by balancing architecture and training decisions.

Similar resource questions exist during post-training finetuning of LLMs. We assume a common setting where a single-base LLM needs to be finetuned for a downstream task. Numerous works have sought to induce certain abilities by training from large instruction tuning datasets (Sanh et al., 2021; Wei et al., 2021; Mishra et al., 2021). There are several different resource constraints which might make this challenging. For example, parameter-efficient fine-tuning methods like LoRA (Hu et al., 2021) aim to reduce memory usage during finetuning by updating only a small subset of the model's parameters. In this work we focus instead on compute-constrained finetuning.

A promising approach to reducing compute requirements for finetuning is *data selection*. Data selection is a foundational approach in machine learning where the objective is to create a minimal dataset from a collection of data (Hart, 1968; John, 1975). Given the large computational cost of each gradient step, reducing dataset size is an appealing way to reduce this resource usage. Moreover, as larger and more diverse instruction tuning collections become available, likely only a subset of the data provides the most value for any given task. Recent work has shown that careful data selection can vastly increase the effectiveness of finetuning per step (Chen et al., 2023; Zhou et al., 2024).

Yet, even if data selection is effective, it does not *a priori* imply that it is compute-optimal. Given the base-level compute effectiveness of gradient descent on LLM models, data selection methods need to improve upon standard training in proportion to their added cost. In other words, a compute-optimal method should both improve training and be cheap to compute. In this work, we study this setting of compute-constrained data selection, and argue that this is a critical factor for practical adoption that is being under considered in method development.

Concretely, we aim to quantify the trade off between model size, number of tokens, and data selection in LLM finetuning, such that practitioners can make well-informed decisions when choosing how to best allocate compute. We first formalize this problem as a compute-constrained combi-

natorial optimization problem and then discuss our compute-aware modification. We develop an categorization of different approaches for this task and model their compute scaling. This is used to frame a broad series of empirical training runs under varying compute constraints, scaling both model sizes and training length. We train over 600 models, ranging from 7 to 70 billion parameters, across 6 data selection methods and 3 downstream tasks, recording final task performances for each.

The results from this study argue that complex data selection methods are almost never Pareto-optimal in the compute-constrained setting and that simple statistical methods such as sparse retrieval should be preferred. In particular, powerful data selection methods that use model perplexity or gradient information tend to be FLOP inefficient both from a theoretical and empirical perspective. That is not to say these methods are ineffective; for example, they should be used in settings with repeated training with different tasks on the same underlying models. We fit parameteric models to quantify the effectiveness of various approaches in a compute-aware manner. Using these fits, we extrapolate and empirically validate that, for comptue-optimal finetuning, perplexity and gradient data selection require training-to-selection model size ratios of 5x and 10x, respectively.

We hope that this framework and setting can motivate further research into cheaper data selection methods that can produce better models with less compute. Codebase and datasets to reproduce our results are available at https://github.com/oseyosey/CCDS.

## 2 RELATED WORK

**Compute Scaling for Model Size and Transfer Learning.** Kaplan et al. (2020) established the use of compute-scaling laws for language models and showed a power-law relationship between model size and loss over varying orders of magnitude. More recent works expand on the original formulations by considering learning rate schedule matching, multiple-epoch training, and hyperparameters (Hoffmann et al., 2022; Muennighoff et al., 2024; Bi et al., 2024).

Hernandez et al. (2021) study scaling in the post-training setting but only model the relationship between pretraining and finetuning data loss. Similar work (Lin et al., 2024; Isik et al., 2024; Zhang et al., 2024) studies the scaling in post-training by modeling the relation between pretraining data size, model size, finetuning method and downstream test loss. These models do not consider the use of data selection. Recent work considers the relationship between data selection and scaling in the vision domain (Goyal et al., 2024), but this work does not consider the compute needed for data selection in their scaling analysis.

**Data Selection for Language Models.** Data selection takes the full training data as input and chooses a subset to train (Albalak et al., 2024). It can be viewed as a coreset selection problem (Mirzasoleiman et al., 2020; Killamsetty et al., 2021b), which aims to select a subset from the given training dataset such that the model achieves performance similar to the full dataset.

There are many different approaches to LLM data selection. The simplest are non-model specific approaches such as manual scoring functions (Chen et al., 2024), surface level features (Robertson et al., 2009), and n-gram features (Xie et al., 2023). On the other hand, more effective methods use LLMs to assign utility. One class uses model forward inference information such as utility scores from generations, model embeddings, and perplexity (Wettig et al., 2024; Marion et al., 2023; Ivison et al., 2022). Another class uses model gradients to define influence function style selections (Killamsetty et al., 2021a; Han, 2023; Xia et al., 2024). See Section 4 for further description.

**Task-Specific Finetuning from General-Purpose Instruction Datasets.** While data selection can be applied in any finetuning setting, it is most impactful as a method to train a targeted model from a general-purpose dataset. In the case of LLMs, this setting is commonly training a task-specific model from an "instruction-tuning" dataset (Sanh et al., 2021; Wei et al., 2021; Mishra et al., 2021).

Instruction-tuned models are effective for many downstream tasks; however they require training on a very large and expensive set of data. To get around this issue, models have demonstrated strong results using small subsets of instruction tuning data (Chen et al., 2023; Lu et al., 2023; Zhou et al., 2024). As datasets grow, automated measures of quality selection has become a growing focus, particularly when many targeted models are needed. Therefore while instruction-tuning is not the direct focus of this work, it provide a real-world applications of compute-constrained data selection.

## 3 BACKGROUND

The goal of data selection is to choose a subset of data points from a large dataset to optimize model performance on a target task. In a learning task, we are given a large training set, $\mathcal{D}$, a target test dataset, $\mathcal{T}$, and a validation set, $\mathcal{V}$. Our goal is to find the optimal subset $\mathcal{S} \subseteq \mathcal{D}$ such that the model $\theta = T(\mathcal{S})$ trained on $\mathcal{S}$ maximizes the performance on $\mathcal{T}$ under a given data constraint:

$$\mathcal{S}^* = \underset{\mathcal{S} \subseteq \mathcal{D}}{\arg\max} \quad P(\mathcal{T}; T(\mathcal{S}))$$
$$\text{subject to} \quad |\mathcal{S}| \leq K, \tag{1}$$

where $P$ denotes the performance of the model on the test set and $K$ is the max cardinality.

This problem is challenging to solve in the general case, particularly without access to the test set $\mathcal{T}$. Approaches to the problem therefore commonly make two implicit assumptions: (1) the performance function, $P(\mathcal{T}; T(\mathcal{S}))$, is monotonic and submodular, non-increasing marginal utility, in the dataset chosen (Kirchhoff & Bilmes, 2014), and (2) the validation set $\mathcal{V}$ is IID with the test set $\mathcal{T}$. Under these assumptions, we can argue for a greedy data selection approach (Kirchhoff & Bilmes, 2014). This allows decomposing the total objective, $P(\mathcal{T}; T(\mathcal{S}))$, by considering the contribution of individual training points to the performance on the validation set $P(\mathcal{V}; T(\{x\}))$ for each $x \in \mathcal{S}$.

To estimate the marginal contribution of $x \in \mathcal{S}$, most data selection methods use a *utility function* $v(x; \mathcal{V})$—as a proxy to $P(\mathcal{V}; T(\{x\}))$—to give the utility of each data point $x$ based on its relevance (Albalak et al., 2024). By ranking the data points $\mathcal{D}$ based on $v$ and selecting those that maximize the total utility within the data budget $K$, greedy data selection aims to obtain a high-performing subset $\mathcal{S}^*$. To summarize, we consider data selection methods that target Equation (1) with a two-step greedy algorithm: score all points and then select points up to the budget $K$.

## 4 COMPUTE-CONSTRAINED DATA SELECTION

While the framework presented in Section 3 provides a general method for data selection, we argue that it is insufficient for the practical challenge of finetuning LLMs. The issue is that LLM finetuning is often bottlenecked by a computational budget and not a data budget. There are two major computational bottlenecks in this process: (1) the cost of training the model on this data ($C_T$), and (2) the cost of computing the utility function on this data ($C_v$). The true cost of $C_v$ can reduce significantly the amount of training points we can select for given computational budget .

Assuming we at minimum require the computation of a utility function over the dataset, we can define the *compute-constrained data selection* objective as

$$\mathcal{S}^* = \underset{\mathcal{S} \subseteq \mathcal{D}}{\arg\max} \quad P(\mathcal{V}; T(\mathcal{S}))$$
$$\text{subject to} \quad C_{T(\mathcal{S})} + \sum_{x \in \mathcal{D}} C_{v(x)} \leq K. \tag{2}$$

Here $K$ is now the compute, e.g. maximum number of FLOPs, allocated for data selection and training, and we assume calculation of $v$ is a fixed-cost independent of optimization.

### 4.1 COMPUTE COST OF DATA SELECTION UTILITIES

To make these costs more tangible, we consider four classes of data selection in this work, that represent three different levels of compute. This section summarizes their main properties, i.e. their core utility functions and computational costs.

**Lexicon-Based.** Lexicon data selection methods utilize statistical properties of text to evaluate the relevance of data points without relying on deep learning models. One of the most effective lexicon-based methods is *BM25* (Robertson et al., 2009; Silva & Barbosa, 2024), which scores data points based on the frequency of terms. The utility function $v_{\text{BM25}}(x; \mathcal{V})$ assigns a relevance score to each

| Method | Utility Function | Computational Cost | $C_{\mathbf{forward}}(\mathbf{x})$ |
|---|---|---|---|
| Lexicon-Based | $\frac{1}{|\mathcal{V}|}\sum_{x'} \text{BM25}(x, x')$ | $c_{\text{BM25}}(|x| + |\mathcal{V}||x|)$ | $\approx 0$ |
| Embedding-Based | $\frac{1}{|\mathcal{V}|}\sum_{x'} \cos(\text{Emb}(x), \text{Emb}(x'))$ | $C_{\text{embed}}(x) + C_{\text{embed}}(\mathcal{V})$ | $\approx \epsilon$ |
| Perplexity-Based | $\text{PPL}_{\theta_{\mathcal{V}}}(x)$ | $C_{\text{forward}}(x)$ | $\approx 1$ |
| Gradient-Based | $\eta_t \langle \nabla_\theta \ell(x; \theta^t), \nabla_\theta \ell(\mathcal{V}; \theta^t) \rangle$ | $3 \times C_{\text{forward}}(x) + C_{\text{grad}}(\mathcal{V})$ | $\approx 3$ |

Table 1: **Utility Functions and Computational Costs for Data Selection Methods.**

data point by averaging the BM25 scores with the validation, $v_{\text{BM25}}(x) = \frac{1}{|\mathcal{V}|}\sum_{x'\in\mathcal{V}} \text{BM25}(x, x')$. Since the algorithm can be run with a single-core cpu, the data selection FLOPs are almost 0.

**Embedding-Based.** These methods utilize embedding models to select data points that are most similar to the target data (Rubin et al., 2021). The utility function $v_{\text{retrieval}}(x; \mathcal{V})$ assigns a score to each data point $x$ based its cosine similarity with validation data in $\mathcal{V}$; that is, $v_{\text{retrieval}}(x) = \frac{1}{|\mathcal{V}|}\sum_{x'\in\mathcal{V}} \cos(\text{Emb}(x), \text{Emb}(x'))$. Assuming we are using a very small model (on the order of BERT-size) and the embedding requires only one-time transformation of data points into dense vectors, the data selection FLOPs are quite small. Liu et al. (2021) demonstrated significant performance gains when selecting in-context examples. We apply a similar method, *Embed*, to the fine-tuning setting using a small T5-based dense embedding model (Ni et al., 2021).

**Perplexity-Based.** Perplexity-based data selection utilizes language models to evaluate the utility of data points based on model loss (Antonello et al., 2020). The utility function $v_{\text{ppl}}(x; \mathcal{V})$ assigns a score to each data point $x$ by computing the perplexity (PPL) of $x$ under a language model $\theta_{\mathcal{V}}$ finetuned on $\mathcal{V}$; that is, $v_{\text{ppl}}(x) = \text{PPL}_{\theta_{\mathcal{V}}}(x)$. *Top-PPL* and *Mid-PPL* have both shown improved performance and training efficiency (Ankner et al., 2024; Marion et al., 2023), where *Top-PPL* ranks data points with the highest perplexity scores, and *Mid-PPL* does the same for points in the middle of the score distribution.

**Gradient-Based.** These methods evaluate the utility of data points based on their influence on the model's loss with respect to the target data (Pruthi et al., 2020). The utility function $v_{\text{grad}}(x; \mathcal{V})$ quantifies this influence by computing the inner product between the gradient of the loss on $x$ and the gradient of the loss on $\mathcal{V}$, scaled by the learning rate $\eta_t$; that is, $v_{\text{grad}}(x) = \eta_t \langle \nabla_\theta \ell(x; \theta^t), \nabla_\theta \ell(\mathcal{V}; \theta^t) \rangle$, where $\nabla_\theta \ell(\mathcal{V}; \theta^t) = \frac{1}{|\mathcal{V}|}\sum_{x'\in\mathcal{V}} \nabla_\theta \ell(x'; \theta^t)$, and $\theta^t$ are the model parameters at time step $t$. The computational cost of computing $v_{\text{grad}}(x)$ for each $x$ is approximated by:

$$C_{v_{\text{grad}}}(x) \approx C_{\text{backward}}(x) + C_{\text{grad}}(\mathcal{V}) \approx 3 \times C_{\text{forward}}(x) + C_{\text{grad}}(\mathcal{V}),$$

where $C_{\text{forward}}(x)$ is the cost of a forward pass for $x^{(i)}$ on the model $\mathcal{M}$, and $C_{\text{grad}}(\mathcal{V})$ is the cost of computing $\nabla_\theta \ell(\mathcal{V}; \theta^t)$, computed once for all $x$. Computing gradients involves both forward and backward passes, totaling approximately three times the cost of a forward pass. Low-rank sgradiEnt Similarity Search (*LESS*) shows superior performance gain over cheaper methods and random selection (Xia et al., 2024).

While lexicon and embedding-based methods aim to select training samples similar to validation samples, perplexity and gradient-based methods focus on optimizing their effect on model loss. Different implementation of these general categories leads to different cost. A detailed cost analysis of our selected data selection methods can be found in the Appendix B.

## 5 MODELING THE COMPUTE-PERFORMANCE RELATIONSHIP

To analyze the trade-off between the compute of data selection methods and the expected gain in model performance, we define a simplified parametric form for expected peformance. Let $c$ be the fixed-cost of training on a single data point and $k = |\mathcal{S}|$ be the number of data points. Define $C(k)$ as the total cost of training and selection,

$$C(k) = c \times k + \sum_x C_{v(x)}.$$

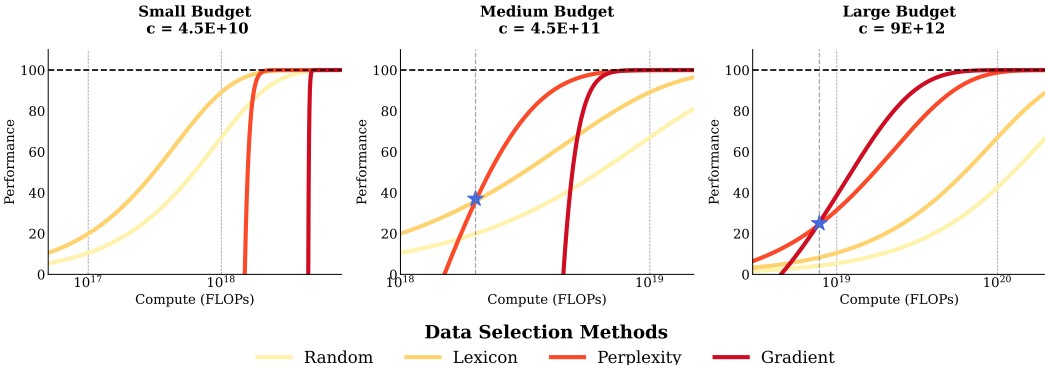

Figure 1: **Simulation of Performance under Constraints.** $P(k) = \bar{P} \times \left(1 - \exp\left(-\lambda \frac{C(k)}{C(|\mathcal{D}|)}\right)\right)$
The behavior of different data selection methods using our performance model. *(Left-Small Budget)* The Lexicon method may consistently outperform more advanced data selection methods if their initial cost is too high. Under our assumptions gradient can never be optimal as its cost exceeds 1 epoch of training. *(Middle-Medium Budget)* The perplexity method can become optimal once the total cost exceeds a given amount. *(Right-Large Budget)* The gradient methods can be optimal if training is more expensive than the fixed-cost, for example if using a much larger base model than data selection model. The simulation shows that the compute-optimal data selection method changes as a function of the compute budget and the performance rate associated with each method.

The assumption in data selection is that more compute intensive methods are able to achieve higher performance with less samples, but that the value of information gained from selecting additional data points diminishes as more points are explored. We will assume that data points are exchangeable, and that all methods eventually reach the same upper-bound performance after 1 epoch, $\bar{P} = P(\mathcal{T}; T(\mathcal{D}))$.

Our parametric model for the expected performance after training on $k$ data points should capture diminishing returns, dependence on computational cost of the method, and convergence toward an upper bound. We model this function as:

$$P(k) = (\bar{P} - P_0) \times \left(1 - \exp\left(-\lambda \frac{C(k)}{C(|\mathcal{D}|)}\right)\right) + P_0 \qquad (3)$$

where $P_0$ is the zero-shot performance, $\bar{P}$ is the upper bound and $\lambda$ is the value the method extracts from additional compute in the utility function.

**Simulation**   Figure 1 presents a simulation of this performance function across different methods under varying compute constraints. We set $c$ to $\{4.5E+10, 4.5E+11, 9E+12\}$, $k$ to 100M, and utility compute to $\{4.5E+10, 4.5E+11, 9E+12\}$. The parameter $\lambda$ is set to $\{5, 10, 40, 80\}$ for the Random, Lexicon, Perplexity, and Gradient methods, respectively. These settings demonstrate how different data selection methods may become compute-optimal under varying compute budgets. Note that for Medium and Large Budgets, we use a Small-size model for PPL and Gradient.

Using methodology similar to Hoffmann et al. (2022), the parameter $\lambda$ can be fit on the empirical measurements, which yields data-driven estimates. See Appendix C.1 for more details on derivations and the fitting procedure.

## 6   EXPERIMENTAL SETUP

As shown in Table 2, experiments vary the number of finetuning tokens for 5 data-selection methods and a fixed family of models, ranging from 7B to 70B parameters. The finetuning data budget is fixed as a percentage of the total finetuning tokens: $\{2.5, 5, 10, 25, 50, 100\}\%$, across 3 target tasks. For each finetuning budget, we conduct multiple training runs with increasing compute, which is either allocated toward larger pre-trained model sizes or more sophisticated data selection methods.

| Training Data | Data Selection Method | Model Size | Target Task |
|---|---|---|---|
| 2.5% | Random | LLAMA2 7B | MMLU |
| 5% | BM25 | LLAMA3 8B | BBH |
| 10% | Embed | LLAMA2 13B | IFEval |
| 25% | PPL | LLAMA2 70B | |
| 50% | LESS | | |
| 100% | | | |

Table 2: **Experimental Setup Overview**.

We analyze each FLOP count to identify which runs achieve the highest performance on target benchmark. We then fit a power law to obtain a finetuned Pareto frontier for each model sizes.

**Datasets**  We follow Wang et al. (2023) and curate a representative sample of instruction-tuned datasets as listed in Table 8. This includes: (1) datasets generated by researchers from existing NLP datasets, such as COT (Wei et al., 2022) and Flan V2 (Longpre et al., 2023) ; (2) datasets written by humans from scratch specifically for instruction tuning, including Dolly (Conover et al., 2023) and Open Assistant 1 (Köpf et al., 2024).

For evaluating the model we run on three challenging but different downstream tasks. These include: the Massive Multitask Language Understanding (**MMLU**, Hendrycks et al. (2020)) dataset measures models' factual knowledge, comprising questions across 57 subjects, spanning difficulty levels from elementary to professional; Big-Bench-Hard (**BBH**, Suzgun et al. (2022)) curates 23 complex reasoning tasks from Big-Bench Srivastava et al. (2022). It is used to evaluate models' general reasoning capabilities; and Instruction Following Evaluation (**IFEval**, Zhou et al. (2023)), which evaluates models' instruction following abilities. For MMLU, we report 5-shot accuracy; for BBH, we report 3-shot exact match score; and for IFEval, we report 0-shot accuracy.

**Pretrained Models**  For all experiments, we train transformer language models with LLAMA architecture and tokenizer (Touvron et al., 2023; Dubey et al., 2024). Models range from 7B parameters to 70B parameters, and are trained for close to 100 million total fine-tuning tokens. Our experiments closely follow prior work on training and evaluating instruction-tuned models (Wang et al., 2023; Ivison et al., 2023).

We primarily focus on the LLAMA-2 model suite, containing LLAMA-2-7B, LLAMA-2-13B, and LLAMA-2-70B. Because the latest LLAMA-3 model suit only contains two model sizes (8B and 70B), we believe the three LLAMA-2 models are better suited in modelling the scaling behaviour. Nevertheless, We show that our results generalizes well to LLAMA-3 by experimenting with the smaller base model LLAMA-3-8B.

## 7  RESULTS

**Empirical Results.**  Figure 2 shows the full results with 5 data selection methods across 3 pretrained model size and 2 of the target task. For these experiments, the unique training data budget is fixed at roughly {2.5%, 5%, 10%, 25%, 50%, 100%} of tokens. For each data budget, we finetune a set of models with increasing amount of compute that is allocated to either more parameters or more expensive data selection methods.

Note that for PPL and Gradient, a 7B model of the same model family is always used for data selection; whereas for Embed a small encoder model is used. For MMLU, PPL is implemented as *Mid-PPL*. For BBH, PPL is implemented as *Top-PPL*.

*The fine-tuned efficient Pareto frontier* comprises all runs that are Pareto-optimal with respect to compute (x-axis) and performance (y-axis). These runs represent the most efficient choices, providing the best possible performance for a given compute budget under specific data selection methods and training token lengths. Furthermore, we model the efficient computational frontier using a power law, specifically of the form $P(C) = a \log(C) + b$, where $a$ and $b$ are parameters fitted to the data. The fitted function is depicted as a dashed gray line in the figures.

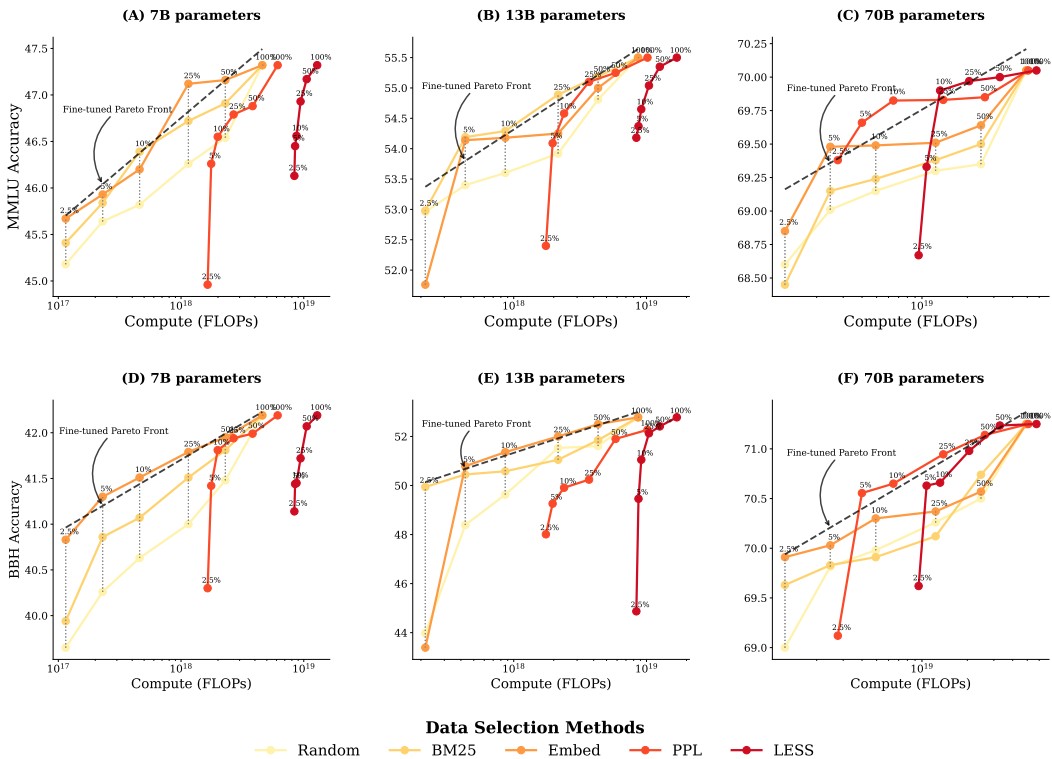

Figure 2: **Performance for Different Data Selection Methods**. We show all of our different runs for a given model size, where each scatter point is the final target task performance of a single run. *(A, B, C)* show MMLU results across three model sizes, while *(D, E, F)* present BBH results across three model sizes. For each run, we determine the optimal finetuning strategy—a combination of data selection method and number of finetuning tokens—that achieves the highest performance under a particular FLOPs budget. We fit a pareto front in dashed line based on these optimal strategies, which is a line in the linear-log space. At small and medium compute budgets *(A, B, D, E)*, cheaper data selection methods like BM25 and EMBED outperform PPL and LESS, which rely on model information. At larger compute budgets *(C, F)*, however, PPL and LESS become compute-optimal after using 5% of the fine-tuning tokens.

The main 7B results in Figure 2 *(A, D)* and Figure 5 (b), show that cheap lexicon-based methods (BM25) and embedding-based methods (Embed) significantly outperform perplexity-based (PPL) and gradient-based methods (LESS). While PPL and LESS achieve better performance at the same *data budget* compared to these methods (see Figure 5), they are not compute optimal under the same *compute budget* due to the high FLOPs required for data selection. The marginal benefit one can get from using a more sophisticated data selection methods does not outweigh its cost in selecting these data. Additional results can be found in Appendix F.

As models scale to 13B Figure 2 *(B, E)*, we still find that expensive data selection methods underperform, even though, the relative cost of doing data selection for these methods diminishes as the training model size increases. As shown in Figure 2 *(B, E)*, at 13B model sizes, cheaper methods are still preferred. We do see that after 5% finetuning tokens, PPL is more competitive, outperforming Embed in MMLU, and almost matching the pareto front at 25% finetuning tokens.

At the 70B model size, shown in Figure 2 *(C, F)*, PPL and LESS outperform both BM25 and Embed finetuning tokens for the first time. This suggests that at very large compute budgets, more sophisticated and costly methods can gain a greater advantage compared to lexical and embedding methods. As model sizes continue to scale and LLM sizes shrink, these methods may become more efficient.

Results on LLAMA3 8B are nearly identical to the 7B we see for LLAMA2. This verify that the approach is not model specific. We include full results in Appendix E.

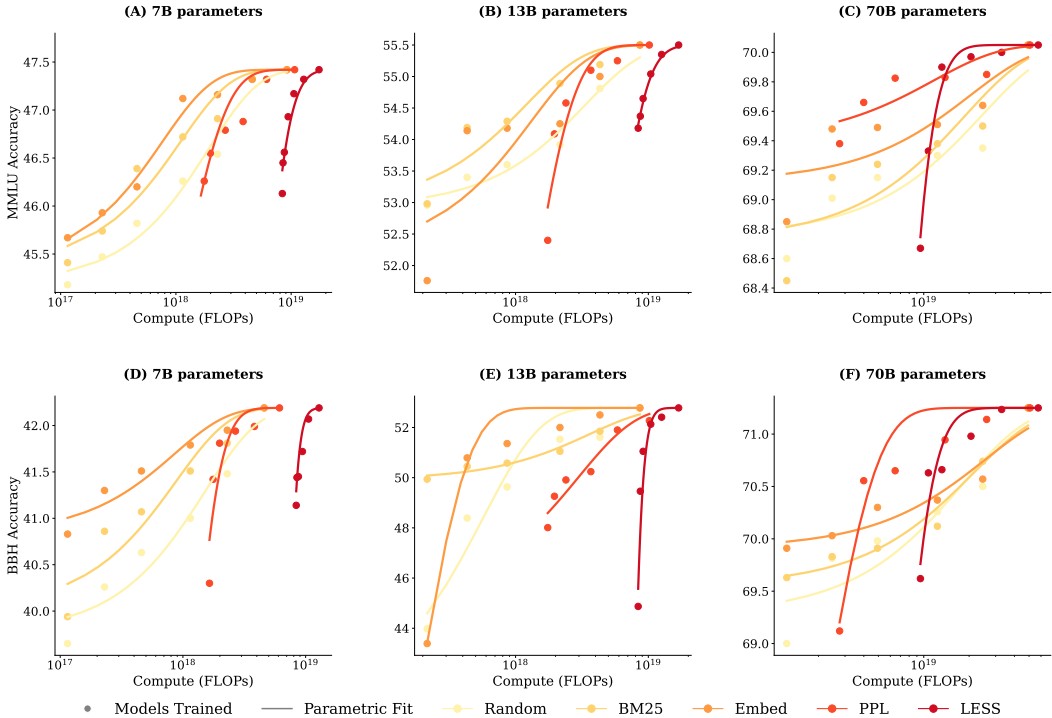

Figure 3: **Parametric Fit of Performance with Compute-Constrained Data Selection.** We fit a parametric model of the performance in Equation (3) and display that as curves to pair with the empirical results as scatter points. *(A, B, C)* show MMLU results and their parametric fit across three model sizes, while *(D, E, F)* present BBH results and their parametric fit across three model sizes.

**Fit of Compute-Performance Relationship.** In Section 5 we propose a parameteric model for the relationship of data selection compute to model performance, Equation (3). This formula assumed that data selection diminish in utility and that performance is a direct function of compute. To verify these assumption, we fit the parametic form to the the empirical curves by fitting the $\lambda$ parameter and allowing tolerance in $P_0, \bar{P}$, per method and dataset. We model all final performances from our experiments as a parametric function of the compute budget $P(k)$.

**Extrapolation from Parametric Fits.** Figure 3 shows the fitted curves for each method. Since the shape of these curves is dataset and model dependent, we cannot predict exact performance for different levels of data selection compute. However, the close fit of the parametric models obtained from smaller models allows us to estimate the compute-optimal ratio between the training model size and the selection model size. For perplexity-based data selection, our extrapolation suggests that the method becomes compute-optimal when the training model is 5x larger than the data selection model—around 35B parameters. For gradient-based data selection, our extrapolation indicates that the training model needs to be approximately 10x larger than the data selection model to be compute-optimal—around 70B parameters. Appendix G presents our results and details the extrapolation.

## 8   ANALYSIS

**Comparing Training versus Total Compute Budgeting.**   While our primary interest is in the full compute constrained setting, we note that different results hold if targeting only a small training budget as in Equation (1). When the training-budget is fixed, we observe in  Figure 5 (a) that the gradient-based method (LESS) consistently outperforms other data selection methods, with the perplexity-based method (PPL) being the closest competitor.

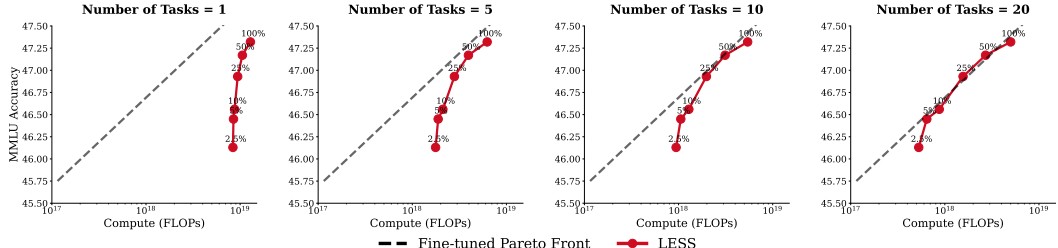

Figure 4: **Multiple Task-Specific Model Break-Even Analysis** . Costs to perform gradient-based method (LESS) are spread over all the target tasks. Performance under compute-constraints reach the finetuned Pareto frontier at 10 tasks, surpassing it at 20 tasks.

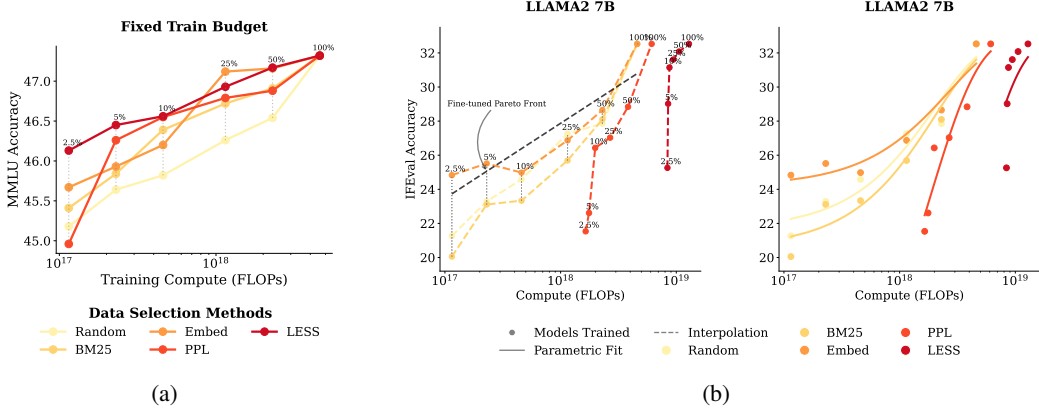

(a)                                    (b)

Figure 5: (a) **Fixed Training Budget.** Considering only training budget, sophisticated methods consistently outperforms cheaper methods. (b) **Performance and Parametric Fit** on IFEval. At small compute budget, sophisticated methods are not compute optimal.

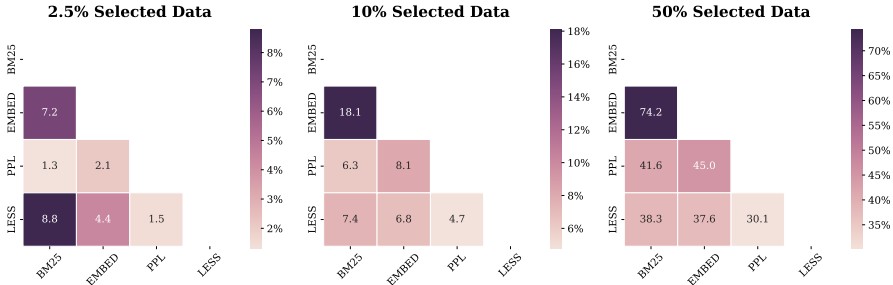

Figure 6: **Data Similarity Between Data Selection Methods** for MMLU.

**Multiple Task-Specific Models.** Another interesting setting is when the goal is to train multiple models from the same large training set, where each targets a different task. In this setting, the gradient information calculation can be amortized between target tasks. However, even in this setting the cost of gradient calculation is still quite high. As shown in Figure 4, with LLAMA2 7B as the data selection model, LESS needs 20 tasks to surpass the finetuned pareto frontier. We perform similar analysis for other model sizes in Appendix H.

**Data Similarity.** Figure 6 presents the Jaccard similarity of data selected by different data selection methods. We observe that BM25 and Embed tend to select highly similar data, which explains their comparable performance. In contrast, the data selected by LESS bears the least resemblance to the other methods. We perform similar analysis for different target tasks in Appendix I.

## 9 CONCLUSION

Data selection is a valuable tool for fine-tuning language models, but its primary benefit — saving compute — must be balanced against the compute costs required to identify an optimal dataset. In practice, popular methods are surprisingly compute intensive, yielding training data reduction at the cost of worse compute-constrained performance. We hope that these results motivate further research into more compute-efficient data selection methods. As demonstrated by our 70B finetuning experiments, sophisticated data selection methods can leverage smaller models for data selection, allowing larger models from the same family to be trained more efficiently in the LLM setting.

## 10 ACKNOWLEDGMENTS

We would like to thank Woojeong Kim and Celine Lee for their insightful feedback on the paper's writing. We are grateful to Mengzhou Xia for assistance with setting up the codebase and valuable discussions on data selection. Additionally, we thank Junxiong Wang, Jing Nathan Yan, and Wenting Zhao for their support in conducting the early experiments. This work was supported by NSF IIS-1901030 and NSF CAREER 2037519.

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

## A  FLOPs Computation

To compute the training FLOPs of a transformer model, we assume the backward pass has twice the FLOPs of the forward pass and follow the FLOP counts of various components of a transformer model for a single forward pass as in Kaplan et al. (2020) as in Table 3. We account for all training FLOPs in our analysis, including those from the embedding matrices. For large models, the contribution of FLOPs and parameters from embedding matrices is minimal. We apply a factor of 2 to represent the multiply-accumulate cost (MAC).

| Operation | Parameters | FLOPs per Token |
|---|---|---|
| Embed | $n_{\text{vocab}}d_{\text{model}}$ | $2n_{\text{vocab}}d_{\text{model}}$ |
| Attention: QKV | $n_{\text{layer}}d_{\text{model}}3d_{\text{attn}}$ | $2n_{\text{layer}}d_{\text{model}}3d_{\text{attn}}$ |
| Attention: Mask (Q-K Dot) | — | $2n_{\text{layer}}n_{\text{ctx}}d_{\text{attn}}$ |
| Attention: Project | $n_{\text{layer}}d_{\text{attn}}d_{\text{model}}$ | $2n_{\text{layer}}d_{\text{attn}}d_{\text{model}}$ |
| Feedforward | $n_{\text{layer}}3d_{\text{model}}d_{\text{ff}}$ | $2n_{\text{layer}}3d_{\text{model}}d_{\text{ff}}$ |
| De-embed (Head) | $d_{\text{model}}n_{\text{vocab}}$ | $2d_{\text{model}}n_{\text{vocab}}$ |
| Total (Non-Embedding) | $N' = 2d_{\text{model}}n_{\text{layer}}(2d_{\text{attn}} + 1.5d_{\text{ff}}) + d_{\text{model}}n_{\text{vocab}}$ | $C_{\text{forward}} = 2N' + 2n_{\text{layer}}n_{\text{ctx}}d_{\text{attn}}$ |
| Total | $N = 2d_{\text{model}}n_{\text{layer}}(2d_{\text{attn}} + 1.5d_{\text{ff}}) + d_{\text{model}}n_{\text{vocab}} + n_{\text{vocab}}d_{\text{model}}$ | $C_{\text{forward}} = 2N + 2n_{\text{layer}}n_{\text{ctx}}d_{\text{attn}}$ |

Table 3: Parameters and FLOPs per token for different operations.

For better granularity, we show a comparision between our calculation and one that uses the common approximation $C_{train} = 6ND$, where $C_{train}$ denotes FLOPs, $D$ denotes total number of training tokens, and $N$ is the number of parameters in Table 4. In practically, the differences in FLOP calculation is small and thus pose no difference our analysis.

| Model | $N$ | $n_{\text{layer}}$ | $n_{\text{ctx}}$ | $d_{\text{model}}$ | $d_{\text{ff}}$ | $d_{\text{attn}}$ | FLOPs per token | FLOP Ratio (Ours/6ND) |
|---|---|---|---|---|---|---|---|---|
| Llama 2 7B | 7B | 32 | 4096 | 4096 | 11008 | $128 \times 32$ | 4.69e+10 | 1.12 |
| Llama 2 13B | 13B | 40 | 4096 | 5120 | 13824 | $128 \times 40$ | 8.82e+10 | 1.13 |
| Llama 2 70B | 70B | 80 | 4096 | 8192 | 28672 | $128 \times 80$ | 5.03e+11 | 1.20 |

Table 4: FLOP comparison. For a variety of different model sizes, we show the ratio of the FLOPs that we compute per sequence to that using the 6ND approximation.

We use this in our calculation of data selection costs for methods that utilizes LLMs (i.e. perplexity-based and gradient-based).

## B  Data-Selection FLOPs

In this section, we detailed our estimation of the costs to perform different data selection methods.

| Data Selection Method | FLOPs Cost |
|---|---|
| BM25 | $1 \times 10^8$ |
| Embed | $4.4 \times 10^{16}$ |
| PPL | $1.53 \times 10^{18}$ |
| LESS | $8.27 \times 10^{18}$ |

Table 5: Data Selection Cost Summary for Each Data Selection Method

| | **Warmup LoRA Training** | | **Gradient Features Computation** | | **Data Selection** | |
|---|---|---|---|---|---|---|
| **Compute** | **Complexity** | **Actual** | **Complexity** | **Actual** | **Complexity** | **Actual** |
| Compute | $\mathcal{O}(|\mathcal{D}_{\text{warmup}}| \cdot N)$ | 6 Hours | $\mathcal{O}(|\mathcal{D}| \cdot N)$ | 48 Hours | $\mathcal{O}(|\mathcal{D}| \cdot |\mathcal{D}_{\text{val}}| \cdot d)$ | $\leq$ 1 Min |
| Storage | - | - | $\mathcal{O}(|\mathcal{D}| \cdot N \cdot d)$ | 17.7 GB | - | - |

Table 6: Asymptotic complexity, wall-clock runtime (measured as single A100 GPU hours), and storage cost associated with each step in LESS Xia et al. (2024).

### B.1  BM25

We assume that each data point incurs a computational cost of 1 FLOP for BM25, denoted as $c_{\text{BM25}} = 1$. The total computation cost is approximated as proportional to the size of the dataset, $|\mathcal{D}|$. Therefore, the data selection cost for BM25 is estimated to be $1 \times 10^8$ FLOPs.

### B.2  EMBED

For the embedding-based method, we approximate the computation cost using the formula $C_{\text{forward}} = 2ND$, where $N$ represents the number of model parameters and $D$ is the dataset size. Given that the embedding model used has $N = 220M$ parameters (Ni et al., 2021), the data selection cost for Embed is estimated to be $4.4 \times 10^{16}$ FLOPs.

### B.3  PPL

Perplexity-based methods require passing every data point through the language model. Given that the cost per token for a 7B model is $c = 4.69 \times 10^{10}$, as shown in Table 4, the data selection cost for PPL is equivalent to one forward pass. Therefore, using the LLAMA2 model for data selection, we approximate the perplexity computation cost to be $1.53 \times 10^{18}$ FLOPs.

### B.4  LESS

LESS involves a two-step process: a 4-epoch warm-up training on 5% of the dataset $\mathcal{D}$ followed by gradient feature computation over the entire training and validation datasets. First, we calculate the FLOPs required for the 4-epoch warm-up training. Then, using the relationship between the time required for warm-up training and gradient feature computation, as outlined in Table 6 from (Xia et al., 2024), we estimate the FLOPs needed for gradient feature computation. Combining these, we approximate the total cost of LESS using LLAMA2 as the data selection model to be $8.27 \times 10^{18}$ FLOPs.

## C  PARAMETRIC FUNCTION

### C.1  FITTING PARAMETRIC FUNCTION

In this appendix, we describe the process for fitting a parametric model that captures the relationship between the number of data points $k$ and performance $P(k)$, as a function of computational cost. The model captures diminishing returns, dependence on computational cost, and convergence toward an upper bound.

We model the expected performance $P(k)$ after training on $k$ data points as follows:

$$P(k) = (\bar{P} - P_0) \times \left( 1 - \exp\left( -\lambda \frac{C(k)}{C(|\mathcal{D}|)} \right) \right) + P_0$$

where:

- $P_0$ is the zero-shot performance (i.e., performance without additional training).
- $\bar{P}$ is the upper bound on performance (i.e., the maximum achievable performance).

- $\lambda$ is a parameter controlling how efficiently the method extracts value from additional compute.
- $C(k)$ is the computational cost of selecting and training on $k$ data points.
- $C(|\mathcal{D}|)$ is the total computational cost of training on the entire dataset.

The goal is to fit the parameters $P_0$, $\bar{P}$, and $\lambda$ to observed data. We fit the model by minimizing the difference between the predicted performance $P(k)$ and the observed performance $P_{\text{obs}}(k)$. This is formulated as the following optimization problem:

$$\min_{P_0, \bar{P}, \lambda} \sum_{i=1}^{N} \left( P(k_i; P_0, \bar{P}, \lambda) - P_{\text{obs},i} \right)^2$$

where:

- $N$ is the number of data points.
- $P_{\text{obs},i}$ is the observed performance at the $i$-th data point.
- $k_i$ is the number of data points used for training in the $i$-th observation.
- $P(k_i; P_0, \bar{P}, \lambda)$ is the predicted performance using the parametric model.

To ensure meaningful parameter estimates, we impose the following constraints:

- $P_0 \geq 0$, as performance cannot be negative.
- $P_0 \leq \bar{P}$, ensuring that performance does not exceed the upper bound $\bar{P}$.
- $\lambda \geq 0$, as $\lambda$ represents the rate at which performance improves with compute.

The parameter $\bar{P}$ is set slightly above the maximum observed performance:

$$\bar{P} = \max_i P_{\text{obs},i} + \epsilon$$

where $\epsilon = 0.05$ is a small buffer to ensure convergence to the upper bound.

We set the initial guesses for the parameters as follows:

- $P_0^{(0)} = P_{\text{obs},1}$, the observed performance at zero-shot (i.e., without training).
- $\bar{P}^{(0)} = \max_i P_{\text{obs},i}$.
- $\lambda^{(0)} = 1.0$, a reasonable initial guess for the compute extraction efficiency.

We use the Levenberg-Marquardt algorithm to minimize the objective function. This method is effective for solving non-linear least squares problems and efficiently handles the non-linear nature of our parametric model.

The optimization problem is solved as follows:

$$(P_0^*, \bar{P}^*, \lambda^*) = \arg \min_{P_0, \bar{P}, \lambda} \sum_{i=1}^{N} \left( P(k_i; P_0, \bar{P}, \lambda) - P_{\text{obs},i} \right)^2$$

The optimization yields the following fitted parameters:

$$P_0^* = [\text{fitted value}], \quad \bar{P}^* = [\text{fitted value}], \quad \lambda^* = [\text{fitted value}]$$

These fitted parameters provide a close match between the observed data and the model and helps us understand for better decision-making in resource allocation.

# D EXPERIMENTAL DETAILS

## D.1 FINETUNING SETTINGS

All experiments were conducted with parameter-efficient finetuning method LoRA Hu et al. (2021). For the LoRA adapter, we specified a rank of 128, an $\alpha$ value of 512, and a dropout rate of 0.1 and applied it across all attention matrices. Adding the LoRA adapter introduce minimal FLOPs overhead during training—having no impact on our FLOPS analysis—and mainly reduce memory requirements for more accessible training.

We follow standard practices in LLM finetuning Wang et al. (2023); Ivison et al. (2023) and use the AdamW optimizer with beta-parameters $(\beta_1, \beta_2) = (0.9, 0.99)$.

The learning rate is set to 2e-5 for the 7B/8B/13B models and 1e-5 for the 70B models. For data budget {2.5%,5%}, we double the learning rate to ensure convergence in loss. For all experiments, we use a warmup ratio of 0.03, BFloat16 precision, and an effective batch size of 128. For 70B model training, we used QLoRA to reduce the memory requirements and speedup the training.

For smaller data budget (2.5%-5%) experiments, we perform five trials across five random seeds. For larger data budget (10%-100%) experiments, we perform three trials across three random seeds. We report mean target task performance in our analysis. Optimization seeds are controlled through the entirety of the experiments.

## D.2 PRETRAINED MODELS

Table 7 lists the pre-trained models we finetuned in this work. We expect our findings to generalize to these models and future, stronger open base models.

## D.3 TRAINING DATASETS

For training, we use the same four processed datasets as in Wang et al. (2023); Ivison et al. (2023), all of which are either human-authored or annotated. Details are provided in Table 8. The FLAN V2 and COT datasets are derived from existing NLP datasets, while DOLLY and OPEN ASSISTANT 1 contain open-ended generation examples with human-written responses. These datasets vary in format, sequence length, and tasks. Following Wang et al. (2023), we standardize their format using the 'Tulu' structure.

## D.4 EVALUATION DATASETS

We evaluate our method on three benchmark datasets: MMLU (Hendrycks et al., 2020), BBH (Suzgun et al., 2022), and IFEval (Zhou et al., 2023). Table 9 contains more details about each tasks. Each subtask comes with few-shot examples or sample responses, which are used as validation set $\mathcal{V}$ for data selection and as few-shot in-context learning demonstration in evaluation.

An important aspect of the data selection approach is the size of the validation set used for computing utility scores. While a small validation set might cause overfitting or insufficient representation of the target task, our experiments indicate that even modestly sized validation sets (e.g., containing more than 50 examples) are sufficient for methods like BM25 and Embed to perform effectively. Since we are selecting large subsets from the training data, the precision required from the validation set is not overly stringent. These methods can effectively capture relevant training samples without

| Base LMs | # Params | # Tokens |
|----------|----------|----------|
| | 6.7B | 2.0T |
| LLAMA-2 | 13.0B | 2.0T |
| | 65.2B | 2.0T |
| LLAMA-3 | 8B | 15.6T |

Table 7: Base models that we finetuned in this work.

| Dataset | # Instance | Sourced from | # Rounds | Prompt Len. | Completion Len. |
|---------|-----------|--------------|----------|-------------|-----------------|
| FLAN V2 | 100,000 | NLP datasets and human-written instructions | 1 | 355.7 | 31.2 |
| CoT | 100,000 | NLP datasets and human-written CoTs | 1 | 266 | 53.2 |
| DOLLY | 15,011 | Human-written from scratch | 1 | 118.1 | 91.3 |
| OPEN ASSISTANT 1 | 55,668 | Human-written from scratch | 1.6 | 34.8 | 212.5 |

TOTAL NUMBER OF TOKENS: 95.7 MILLION

Table 8: Details of training dataset from Wang et al. (2023). Len. is short for token length.

| Dataset | # Shot | # Tasks | $|\mathcal{V}|$ | $|\mathcal{T}|$ | Answer Type |
|---------|--------|---------|------|------|-------------|
| MMLU | 5 | 57 | 285 | 18,721 | Letter options |
| BBH | 3 | 23 | 69 | 920 | COT and answer |
| IFEval | 1 | - | 50 | 541 | Open Generation |

Table 9: Statistics of evaluation datasets. The selection of evaluation tasks cover different kinds of answer types.

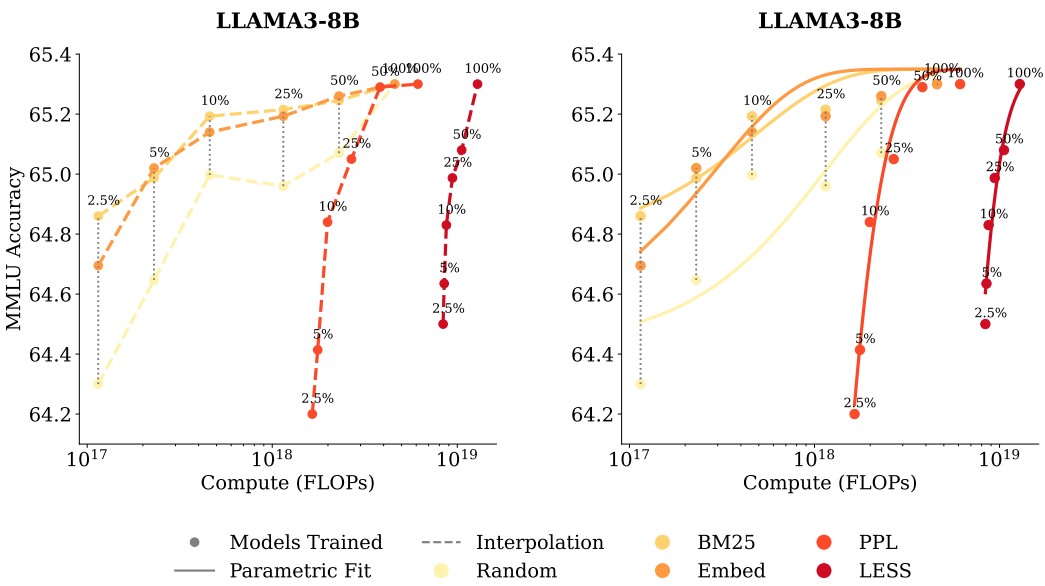

Figure 7: **Performance and Parametric Fit of Performance With Compute-Constrained Data Selection.** *(Left)* We show all of our different runs for a given model size, where each scatter point is the final target task performance of a single run. *(Right)* We fit a parametric model of the performance in Equation (3) and display that as curves to pair with the empirical results as scatter points.

significant risk of overfitting, suggesting that, in practice, reasonably sized validation sets suffice for similarity-based data selection in LLM finetuning.

# E  RESULTS: LLAMA3

We plot additional results on target task MMLU using LLAMA3 8B model in Figure 7. Similar to LLAMA2, LLAMA3 8B results show that cheaper lexicon-based (BM25) and embedding-based (Embed) methods significantly outperform perplexity-based (PPL) and gradient-based (LESS) method. The marginal gains from using more sophisticated methods do not justify their selection costs.

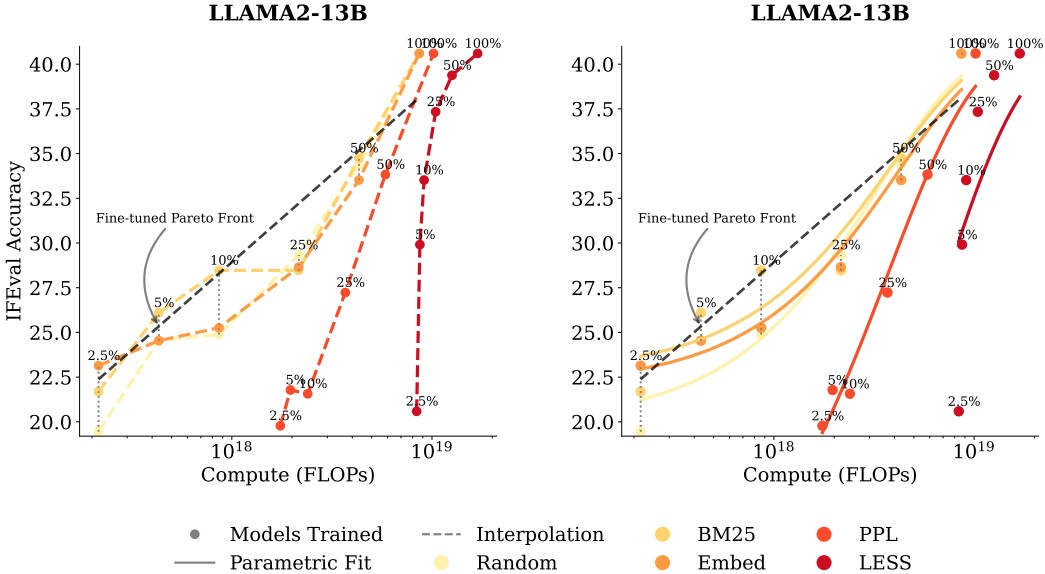

Figure 8: **Performance and Parametric Fit of Performance With Compute-Constrained Data Selection** on IFEval. *(Left)* We show all of our different runs for a given model size, where each scatter point is the final target task performance of a single run. *(Right)* We fit a parametric model of the performance in Equation (3) and display that as curves to pair with the empirical results as scatter points. We fit a pareto front in dashed line based on these optimal strategies. At medium compute budgets, cheaper data selection methods outperform PPL and LESS, which rely on model information.

## F    ADDITIONAL RESULTS: IFEVAL

We plot additional results on target task IFEval with LLAMA3 13B model in Figure 8. As models scale to 13B, expensive data selection methods still underperform, despite their relative cost diminishing with larger models. Cheaper methods remain preferred.

## G    EXTRAPOLATING FROM THE PARAMETRIC FUNCTION

In Figure 9 and Figure 10, we extrapolate the parametric fits obtained from our 7B and 13B model runs by increasing the ratio between the training model size and the data selection model size, thereby effectively reducing the proportional cost of gradient-based data selection. Graphically, this is seen when the parametric fit crosses the existing compute-optimal frontier—a tipping point at which the method becomes compute-optimal.

We found that the training-to-selection model size ratio should be approximately 5 for perplexity-based data selection, and 10 for gradient-based data selection. This suggests that using more expensive data selection methods becomes advantageous when the training model size largely exceeds the selection model size by the compute-optimal ratio. Under these conditions, meaningful efficiency gains can be achieved compared to cheaper data selection methods.

## H    ADDITIONAL ANALYSIS: MULTIPLE TASK-SPECIFIC MODELS.

We perform break-even analysis by varying the number of target tasks for each method. Figure 11 presents the analysis for the 13B model on MMLU, while Figure 12 focuses on the 7B model on BBH and Figure 13 examines the 13B model on BBH. Additionally, Figure 14 and Figure 15 provide results for the 7B and 13B models on IFEval, respectively.

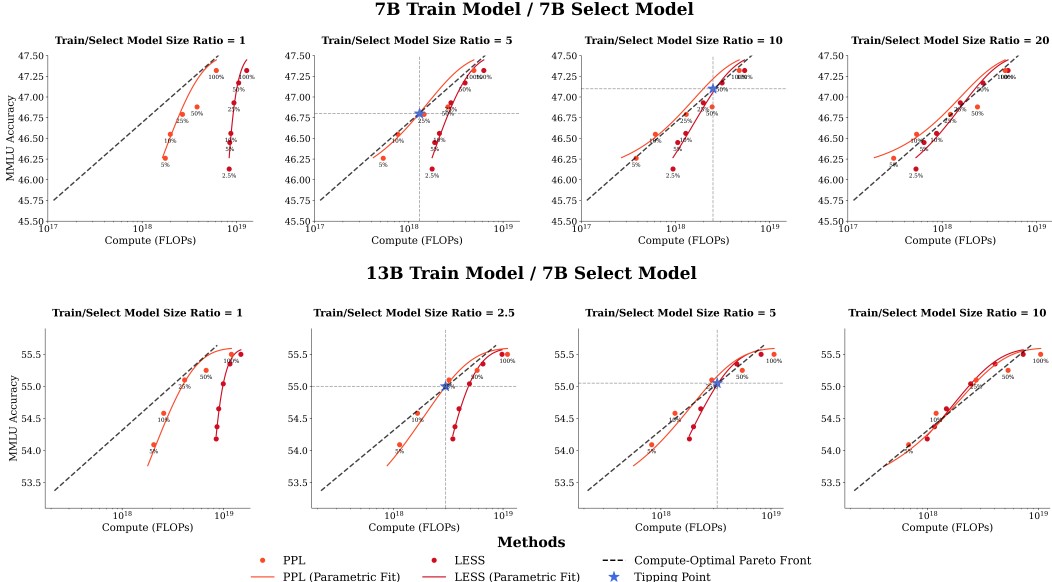

Figure 9: **Extrapolating Compute-Optimal Model Size for MMLU**. By extrapolating from the parametric fits obtained from the 7B and 13B model results for MMLU, we can find the compute-optimal ratio between the training model size and the selector model size required for the perplexity-based and gradient-based method. For perplexity-based data selection, our extrapolation suggests that training model should be larger (5x) than the selector model—around 35B model parameters. For gradient-based data selection to be compute-optimal, our extrapolation suggest that the training model should be significantly larger (10x) than the selector model—specifically, around 70B parameters in this case.

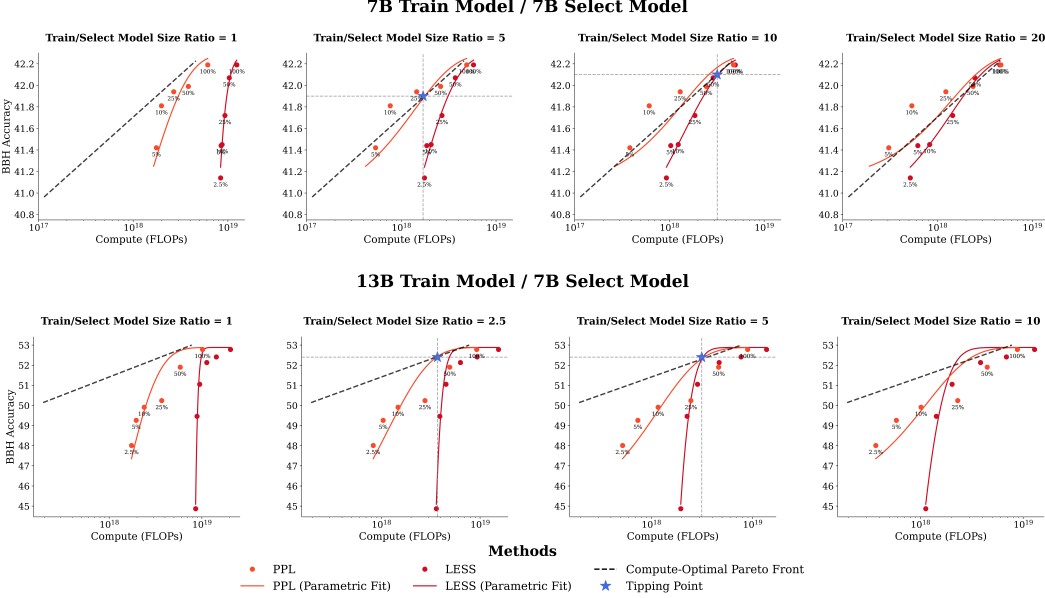

Figure 10: **Extrapolating Compute-Optimal Model Size for BBH**. We perform a similar extrapolation for BBH and find results consistent with those in Figure 9. Specifically, for the perplexity-based method to be compute-optimal, the training model should be 5x larger than the selection model, and for the gradient-based method, the training model should be approximately 10x larger than the selection model.

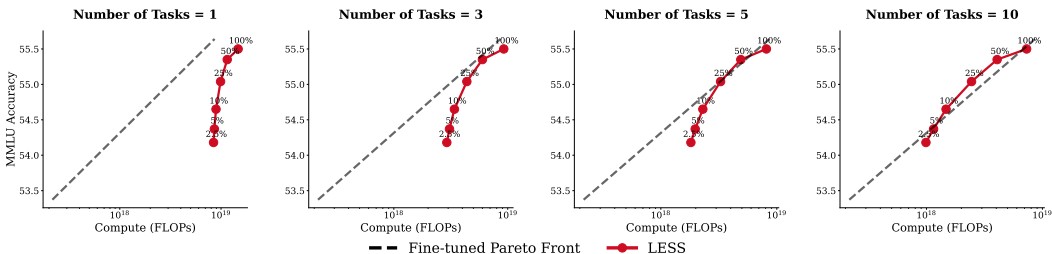

Figure 11: **Multiple Task-Specific Model Break-Even Analysis** for 13B MMLU. Performance under compute-constraints reach the finetuned Pareto frontier at 5 tasks, surpassing it at 10 tasks.

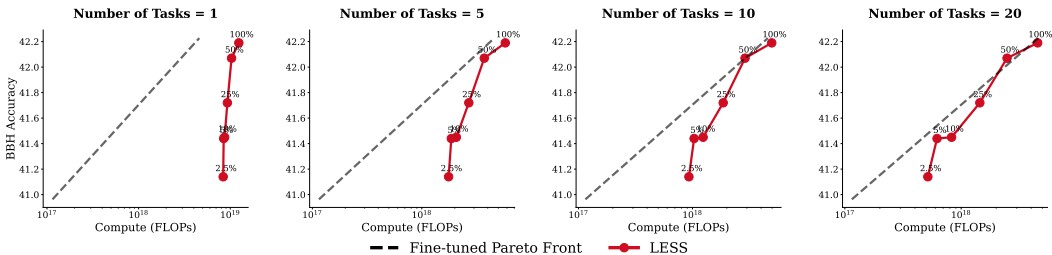

Figure 12: **Multiple Task-Specific Model Break-Even Analysis** for 7B BBH. Performance under compute-constraints reach the finetuned Pareto frontier at 10 tasks.

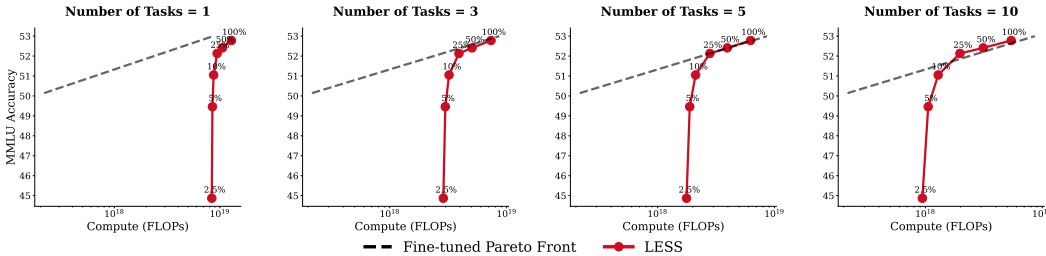

Figure 13: **Multiple Task-Specific Model Break-Even Analysis** for 13B BBH. Performance under compute-constraints reach the finetuned Pareto frontier at 5 tasks, surpassing it at 10 tasks.

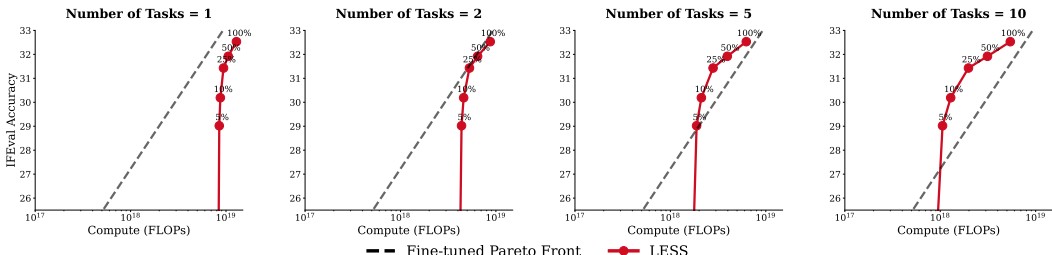

Figure 14: **Multiple Task-Specific Model Break-Even Analysis** for 7B IFEval. Performance under compute-constraints reach the finetuned Pareto frontier at 5 tasks, surpassing it at 10 tasks.

# I  ADDITIONAL ANALYSIS: DATA SIMILARITY

Figure 16 and Figure 17 show the Jaccard similarity of data selected by different data selection methods for the BBH and IFEval target tasks, respectively. Across various percentages of selected data, Embedding and BM25 exhibit the highest similarity. In contrast, LESS shares the least similarity with the other data selection methods.

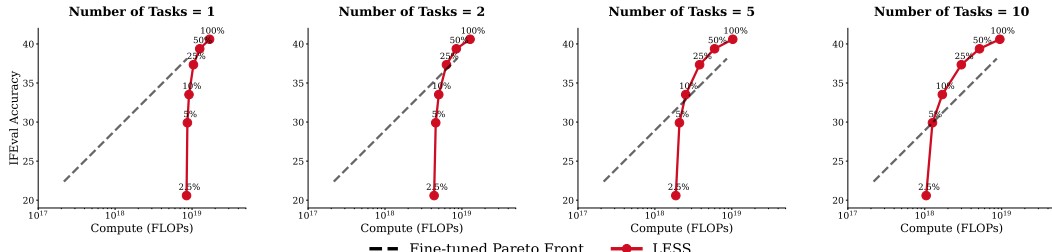

Figure 15: **Multiple Task-Specific Model Break-Even Analysis** for 13B IFEval. Performance under compute-constraints reach the finetuned Pareto frontier at 2 tasks, surpassing it at 5, 10 tasks.

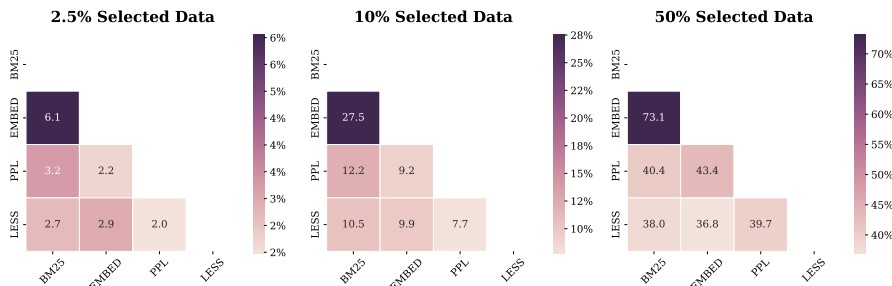

Figure 16: **Data Similarity Between Data Selection Methods** for BBH.

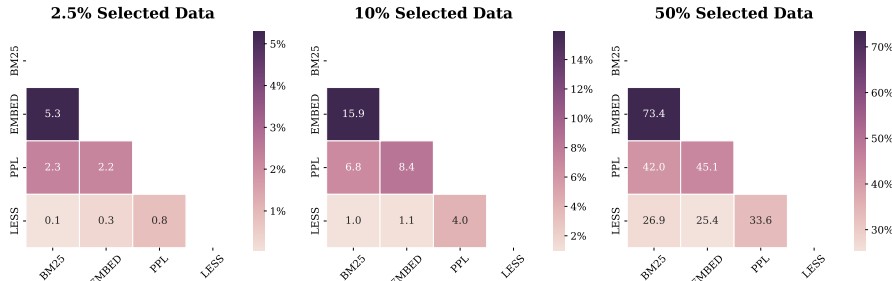

Figure 17: **Data Similarity Between Data Selection Methods** for IFEval.

## J LIMITATIONS

**Repeating Finetuning Data**   In this work, we focus on finetuning the entire dataset for only one epoch. With a larger compute budget, it is possible to repeat fractions or entire datasets multiple times. Multi-epoch settings, as explored in (Xia et al., 2024), could potentially provide further training speedup by repeating data selectively.

**Sensitivity to Hyperparameters**   The effectiveness of finetuning can be highly sensitive to hyperparameters such as learning rate, dropout, or optimizer choice. There may be a specific learning rate that leads to quicker convergence. In this work, we fixed most hyperparameters to commonly used values for fine-tuning LLMs, leaving further exploration of hyperparameter tuning for future work.

**Other Data Selection Methods**   There are additional data selection methods not covered in this work that warrant investigation. While we focused on methods in terms of their compute efficiency, other approaches, such as classifier-based methods, could offer insights and deserve further exploration.

