# OpenReview forum: "Compute-Constrained Data Selection"
_ICLR.cc/2025/Conference — ICLR 2025 Poster_

### Official Review · Reviewer_B9fL · 2024-11-01

**Soundness:** 2
**Presentation:** 3
**Contribution:** 3
**Rating:** 5
**Confidence:** 3

**Summary:**

The authors present a study on compute constrained data selection for training large language models (LLMs).
Unlike preceding works, they do not constrain the size of the training set, but the compute, which is the sum of computational expenditure for data selection as well as LLM training.
They compare various existing data selection methods under this setting and come to the conclusion that many powerful data selection methods are almost never compute-optimal due to their computational cost, making cheaper data selection the favorable choice.

**Strengths:**

- The study is well motivated and its results are of practical importance for finetuning large language models.
- Empirical findings correspond well with theoretical framework

**Weaknesses:**

- The title of the paper is rather broad, while the study is rather specific. "Compute-Constrained Data Selection" does not indicate which type of data is selected for which type of task.

Minor remarks:
- p.3 line 3: \mathcal{S} \subseteq \mathcal{D}, as \mathcal{X} is not introduced
- p. 6 bottom: methods -> method

**Questions:**

- Regarding the notion of utility in Section 3: Is utility here something that is to be minimized, i.e. alternatives with lower utility are preferred over alternatives with higher utility? In the remainder of the paper (expected) model performance is considered for which clearly higher values are preferred.

- I am not sure whether I understood the greedy data selection introduced in Sections 3 and 4. I am under the impression that all data points are scored individually and afterwards they are ranked according to their scores and selected until budget K is exhausted. Isn't it neccesary to do this in an interleaved manner, in order to capture effects like redundancy and the submodularity of utility? Consider the extreme case in which the most informative data point x is repeated K in the dataset, then we would end up with a selection that contains the same datapoint K times.

- In Figure 2, the plot represents performance of Mid-PPL, while the plot in Figure 3 represents performance of Top-PPL. What is the reason for this discreapency?

- In Figure 2, what exactly is the dashed line? Shouldn't the Pareto front contain all solutions, that are dominating on the dimensions of compute (low) and accuracy (high)? The line is straight, is it something like a linear regression applied to the solutions on the Pareto front?

---

> ### Author Response · Authors · 2024-11-21
> **Response to Reviewer B9fL**
>
> We thank the reviewer for the careful feedback. We answer specific questions below:
>
> **Maximize/Minimize Utility**
>
> > Regarding the notion of utility in Section 3: Is utility here something that is to be minimized, i.e. alternatives with lower utility are preferred over alternatives with higher utility? In the remainder of the paper (expected) model performance is considered for which clearly higher values are preferred.
>
> Thank you for the careful read, and we are sorry for the confusion. The utility in Section 3 is intended to be something we maximize. We have since revised Section 3 (see line 113, 130).
>
> **Greedy Data Selection**
>
> > I am not sure whether I understood the greedy data selection introduced in Sections 3 and 4. I am under the impression that all data points are scored individually and afterwards they are ranked according to their scores and selected until budget K is exhausted. Isn't it neccesary to do this in an interleaved manner, in order to capture effects like redundancy and the submodularity of utility? Consider the extreme case in which the most informative data point x is repeated K in the dataset, then we would end up with a selection that contains the same datapoint K times.
>
> You are correct that data selection scores all data points individually based on the utility function $v(x)$, rank them, and select the top K data points until the compute budget is exhausted.
>
> Regarding redundancy and submodularity:
>
> - We assume that each data point $x$ is unique within the dataset $\mathcal{D}$. In practice, the datasets used for fine-tuning LLMs are large and typically undergo preprocessing steps that remove exact duplicates [Wang]. Scenarios where the same data point is repeated $K$ times is negligible in our case.
> - Submodularity decomposes the effects of a selected dataset $S$ into every data points $x \in S$. This implies diminishing returns—adding a redundant or similar data point yields less additional utility [Kirchhoff]. This justifies data selection of scoring data points individually and greedily.
>
>
> **Mid / Top PPL**
>
> > In Figure 2, the plot represents performance of Mid-PPL, while the plot in Figure 3 represents performance of Top-PPL. What is the reason for this discreapency?
>
> Apologies for our mistake. To clarify, MMLU reports performances with Mid-PPL, while BBH uses Top-PPL. We report the best performance for each task from these methods. This has been revised (see line 316–318).
>
> **Pareto Frontier**
> >  In Figure 2, what exactly is the dashed line? Shouldn't the Pareto front contain all solutions, that are dominating on the dimensions of compute (low) and accuracy (high)? The line is straight, is it something like a linear regression applied to the solutions on the Pareto front?
>
> Yes. In our context, Pareto front contains all solutions, that are dominating on the dimensions of compute (x-axis) and performance (y-axis).  These solutions represent the most efficient choices, providing the best possible performance for a given compute budget under specific data selection methods and training token lengths. Furthermore, we assume that the efficient computational frontier can be described by a power-law relationship between the compute budget and number of training tokens. A power law refers to a term of the form $a*log(x)+b$, where terms $a$ and $b$ are fitted. This form have been extensively reference as a form of scaling law in [Kaplan]. We fit this power law to these efficient solution, denoting the fintuned Pareto frontier. This is represented as a line in linear-log space.
>
> We have since updated the paper to clearly define this concept (see Section 7, line 320-324).
>
> *References:*
>
> [Kaplan]: Kaplan, Jared, et al. "Scaling laws for neural language models." *arXiv preprint arXiv:2001.08361* (2020).
>
> [Wang]  Wang, Yizhong, et al. "How far can camels go? exploring the state of instruction tuning on open resources." Advances in Neural Information Processing Systems 36 (2023): 74764-74786.
>
> [Kirchhoff] Kirchhoff, Katrin, and Jeff Bilmes. "Submodularity for data selection in machine translation." *Proceedings of the 2014 Conference on Empirical Methods in Natural Language Processing (EMNLP)*. 2014.

---

> > ### Comment · Reviewer_B9fL · 2024-11-26
> >
> > I would like to thank the authors for the revised manuscript and the response to my questions.
> >
> > I have to say that I am still doubtful regarding the greedy approach that scores the datapoints individually. The approach in (Kirchhoff & Bilmes, 2014) does not score the points individually, as outlined in Algorithm 1 of their paper. The function $f$ scores the candidate points $v$ dependent on the set of already selected points $X_i$. Thus, the algorithm also exhibits quadratic runtime complexity.
> >
> > If the points are first scored individually and afterwards simply ranked and selected until the budget is exhausted, interactions like redundancy cannot be captured.  Needless to say, my example of copying the most informative point $K$ times was a constructed and unrealistic example, but the problem remains: Similar/redundant datapoints would be selected. I think this is a fundamental problem. As none of the other reviewers commented on this, I would like AC to have a look at it and will change my rating from 6 to 5.
> >
> > Apart from the aforementioned point, the authors response was clarifying my questions and I also like the premise of the paper, not introducing a novel method but comparing the currently available methods with a hollistic focus on compute for the overall process of selection and fine tuning.
> >
> > _References_:
> >
> > (Kirchhoff & Bilmes, 2014) Kirchhoff, Katrin, and Jeff Bilmes. "Submodularity for data selection in machine translation." Proceedings of the 2014 Conference on Empirical Methods in Natural Language Processing (EMNLP). 2014.

---

> ### Author Response · Authors · 2024-12-02
> **Response to Reviewer B9fL - Further Discussion on Submodularity & Redundancy**
>
> Thank you for the follow-up. We would like to clarify what we intended to say because we do not think we are in disagreement with your point.
>
> You are correct that the data selection method proposed in Kirchhoff & Bilmes (2014) addresses redundancy, but also is quadratic and unlikely to be tractable in the LLM domain. They note as well that most approaches to this problem do not handle redundancy, stating most data selection methods “value each sentence individually, without regard to any interaction with already selected sentences. This approach, therefore, is **modular (a special case of submodular)** and values a set $X$ via $m(X)$ = $\sum_{x\in X}m(x)$” (Kirchhoff & Bilmes, Section 3).
>
> We mention their work not because we claim to solve redundancy in this but because their use of (sub)modularity provides a general elegant framework for thinking about data selection. **In fact they also note that most practical data select methods rely on (sub)modularity, even when they don't consider redundancy**: “The fact that submodularity is implicit and unintentionally used in previous work suggests that it is natural for this problem” (Kirchhoff & Bilmes).
>
> We agree with you that the modular assumption is not ideal, but it is also the assumption made by all the previous methods benchmarked in this paper. Importantly, this is not an assumption made exclusively in our paper but is common in data selection methods for LLMs due to computational constraints. Given the complexity of these algorithms we do not attempt to handle the general case, but instead are focusing on the modular assumption. They note “the threshold method (greedy algorithm) solves Eqn.2 (core-set selection problem) exactly. On the other hand, a modular function has no chance to represent interaction or redundancy between sentences” (Kirchhoff & Bilmes).
>
> The reviewer acknowledges that our paper presents a unified framework that analyzes existing data selection approaches with a holistic focus on compute—this is the core contribution of our work. While the redundancy assumption is a valid concern, it is not our primary contribution due to practical constraints. We agree though that methods that account for redundancy are an important direction for future work and mentioned (Kirchoff & Bilmes) primarily because we also think handling this efficiently in the future is a critical area of research.
>
> We will add a footnote and discussion urging future work to consider redundancy and submodular formulations.

---

### Official Review · Reviewer_mMXd · 2024-11-03

**Soundness:** 2
**Presentation:** 2
**Contribution:** 1
**Rating:** 5
**Confidence:** 3

**Summary:**

This paper considers selecting data for finetuning LLMs under a computational budget. The computational cost is divided into two parts: 1) when using the validation set to evaluate the performance, the validation set will incur an initial cost; 2) training on each sample will cost a fixed amount of computation. The authors propose an exponential distribution model to fit the model performance v.s. the training costs for four types of data selection methods: lexicon-based, embedding-based, perplexity-based, and gradient-based. The paper consists of numerical experiments over several models and several tasks.

**Strengths:**

1. This paper considers an interesting problem, data selection under computational constraints, and has interesting observations that the initial cost cannot be neglected when considering the computational budget.

**Weaknesses:**

1. (Major) Lack of novelty: although this paper proposes a framework for analyzing the computational cost of each data selection method, it does not provide any new techniques based on this framework. Furthermore, the key observation is not very surprising: the computational cost contains an initial cost when evaluating the validation set, thus the perplexity-based or the gradient-based is clearly not optimal under a limited compute budget.
2. (Major) Lack of soundness: a) the parametric model is selected to be an exponential distribution and the model is fitted to minimize the squared error, but the choice is never justified by any theoretical analysis or numerical results. The fitted curve is also not very convincing (e.g. Figure 3 and Figure 7). b) The Pareto frontier is never formally defined in this paper nor sufficiently discussed. It's very hard for me to believe that the fitted Pareto curve is indeed Pareto optimal as the points in Figure 8 and Figure 10 exceed the Pareto frontier by a large margin. Also, the fitted exponential curves surpass the fitted Pareto curve considerably in Figure 3, implying that the two curves even contradict each other.
3. (Moderate) Lack of insights: this paper is rather an experiment report than a well-motivated paper. The motivation for studying such a computation-constrained data selection problem is not fully supported. The authors just launch a bunch of models, adopt several tasks, and collect all the results without providing sufficient analyses.
4. (Moderate) The style file seems to not follow ICLR 2025 templates: the lines are not numbered.
5. (Minor) Typo(s):
  Figure 1 "using a much larger base model then data selection model": "then" should be "than".

**Questions:**

See the weaknesses part.

---

> ### Author Response · Authors · 2024-11-21
> **Response to Reviewer mMXd (1/4): Novelty**
>
> Thank you for the constructive feedbacks and careful reviews. We provide point-to-point responses to your concern regarding novelty, soundness, and insights.
>
> > (Major) Lack of novelty: although this paper proposes a framework for analyzing the computational cost of each data selection method, it does not provide any new techniques based on this framework. Furthermore, the key observation is not very surprising: the computational cost contains an initial cost when evaluating the validation set, thus the perplexity-based or the gradient-based is clearly not optimal under a limited compute budget.
>
>
> We believe the key observation is more nuanced:
> - While each data selection method incurs an initial cost when evaluating the validation set, **it does not a priori imply that perplexity-based and gradient-based methods are inherently not compute-optimal.** In fact, these methods were developed to enhance training efficiency [LESS; PPL1; PPL2] and are often assumed to provide better performance per unit of compute due to their sophisticated use of model information.
> - To investigate compute-optimal fine-tuning, we formalize this into compute-constrained data selection, showing that compute-optimal data selection is determined by two factors: the cost of the data selection method ($C$) and the rate at which it extracts information from the training dataset ($\lambda$). **Thus, there is no single compute-optimal data selection across all compute budgets; different methods are optimal at different compute budgets.**
> - Our empirical findings reveal that **powerful data selection methods** (perplexity-based and gradient-based) **are less compute-optimal for most practical compute budgets.** Simpler methods like lexicon-based (BM25) and embedding-based (Embed) approaches often outperform them in terms of compute efficiency.
>
> As far as we are aware, this paper is the first to study the optimal scaling properties of data selection for LLM training and show that advanced data selection methods may not be as compute-optimal as they seem.
>
> *References:*
>
> [LESS]: Xia, Mengzhou, et al. "Less: Selecting influential data for targeted instruction tuning." arXiv preprint arXiv:2402.04333 (2024).
>
> [PPL1]: Antonello et al. (2020). Selecting Informative Contexts Improves Language Model Finetuning. arXiv:2005.00175
>
> [PPL2]: Ankner et al. (2024). Perplexed by Perplexity: Perplexity-Based Data Pruning With Small Reference Models. arXiv:2405.20541

---

> ### Author Response · Authors · 2024-11-21
> **Response to Reviewer mMXd (2/4): Soundness**
>
> > The parametric model is selected to be an exponential distribution and the model is fitted to minimize the squared error, but the choice is never justified by any theoretical analysis or numerical results. The fitted curve is also not very convincing (e.g. Figure 3 and Figure 7)
>
> Thanks for the comments, we cover our motivations in Section 5. Below, we outline the reasoning behind selecting an exponential function for our parametric model:
>
> - **Neural Scaling Law:** Scaling laws in LLMs suggest that an exponential increase in FLOPs leads to a linear decrease in loss [Kaplan]. This motivates the use of an exponential function.
> - **Diminishing Returns in Data Utility:** We hypothesize that only a small subset of the data provides the most value for any given task, and that new data points will provide increasingly lower value. The diminishing returns are generally modeled using exponential decay or saturation functions [Muennighoff].
> - **Convergence to an Upper Bound:** We assume that all data selection methods eventually reach the same upper-bound performance after sufficient compute. .
>
> Under these considerations, the exponential saturation formulation we proposed effectively captures the characteristics we want to model: diminishing returns with increasing compute and asymptotic convergence to an upper performance limit.
>
> Importantly, we model performance using downstream task metrics (e.g., MMLU, BBH scores), which are non-linear and not directly proportional to perplexity. Due to this non-linearity, performance can exhibit sharper and less predictable changes as we scale compute [Schaeffer]. We report the average RMSE error of the fitting in the table below; given that the standard deviation of our results is within 0.5, we believe the error is minimal and the fit is reasonable.
>
> **MMLU Fit Losses (RMSE)**
>
> | Model Size | Random | BM25   | Embed  | PPL    | LESS   | Average RMSE |
> | ---------- | ------ | ------ | ------ | ------ | ------ | ------------ |
> | 7B         | 0.1208 | 0.1689 | 0.0991 | 0.3440 | 0.1245 | 0.1715       |
> | 13B        | 0.1534 | 0.2787 | 0.6562 | 0.2923 | 0.0428 | 0.2847       |
> | 70B        | 0.1696 | 0.2191 | 0.1933 | 0.1077 | 0.0734 | 0.1526       |
>
> **BBH Fit Losses (RMSE)**
>
> | Model Size | Random | BM25   | Embed  | PPL    | LESS   | Average RMSE |
> | ---------- | ------ | ------ | ------ | ------ | ------ | ------------ |
> | 7B         | 0.1870 | 0.2313 | 0.1214 | 0.2846 | 0.1243 | 0.1897       |
> | 13B        | 0.8600 | 0.1655 | 0.6348 | 0.3995 | 0.1441 | 0.4408       |
> | 70B        | 0.2511 | 0.1075 | 0.1273 | 0.2353 | 0.1956 | 0.1833       |
>
> > The Pareto frontier is never formally defined in this paper nor sufficiently discussed.
>
> In our context, Pareto front contains all solutions, that are dominating on the dimensions of compute (x-axis) and performance (y-axis).  These solutions represent the most efficient choices, providing the best possible performance for a given compute budget under specific data selection methods and training token lengths. Furthermore, we assume that the efficient computational frontier can be described by a power-law relationship between the compute budget and number of training tokens. A power law refers to a term of the form $a*log(x)+b$, where terms $a$ and $b$ are fitted. This form has been extensively referenced as a scaling law in [Kaplan]. We fit this power law to these efficient solutions, denoting the fine-tuned Pareto frontier. We've updated the paper to clearly define efficient Pareto frontier (see Section 7).
>
> > It's very hard for me to believe that the fitted Pareto curve is indeed Pareto optimal as the points in Figure 8 and Figure 10 exceed the Pareto frontier by a large margin.
>
>
> We apologize for the confusion. To clarify, the break-even analyses in Figures 8 and 10 are exactly performed to examine **how many tasks the gradient-based data selection methods must target to reduce selection costs enough to surpass the current compute-optimal Pareto Frontier**.
>
> > the fitted exponential curves surpass the fitted Pareto curve considerably in Figure 3
>
> The fitted exponential curves are parametric models fitted to the performance data of individual data selection methods, while the Pareto frontier in our analysis is empirically derived. For clarity, we have revised and removed the empirical Pareto front from our parametric fit (see Figure 3).
>
> *References:*
>
> [Kaplan] Kaplan, Jared, et al. "Scaling laws for neural language models." *arXiv preprint arXiv:2001.08361* (2020).
>
> [Schaeffer] Schaeffer, Rylan, Brando Miranda, and Sanmi Koyejo. "Are emergent abilities of large language models a mirage?." *Advances in Neural Information Processing Systems* 36 (2024).
>
> [Muennighoff] Muennighoff, Niklas, et al. "Scaling data-constrained language models." *Advances in Neural Information Processing Systems* 36 (2023): 50358-50376.

---

> ### Author Response · Authors · 2024-11-21
> **Response to Reviewer mMXd (3/4): Insights**
>
> > The authors ... collect all the results without providing sufficient analyses.
>
> We believe our paper goes beyond scaling experiments to provide substantial analyses, including:
>
> - **Parametric Modelling of Performance:** In Section 5, we introduce a parametric model (Equation 3) to quantify the relationship between computational investment and performance gains. We analyze the behavior of different data selection methods theoretically and fit this model to our empirical data.
> - **Compute-Optimality Analysis:** We empirically evaluate the compute-optimality of various data selection methods across model sizes and tasks, finding that more sophisticated methods are often not compute-optimal under practical constraints.
> - **Scaling Extrapolation Analysis:** In Section 7, we analyze how compute-optimal data selection methods vary with model size and compute budget. Our analysis shows that for the complex method to be compute-optimal, the train model size need to 10x the data selection model size.
> - **Break-Even Analysis for Multiple Tasks:** We perform a break-even analysis to identify when the upfront cost of expensive data selection methods is justified over multiple tasks (Section 8 and Appendix H).
> - **Data Similarity Analysis:** We examine the overlap between datasets selected by different methods (Section 8 and Appendix I) to compare data selected by different methods.
>
> If you have specific suggestions for additional analyses, we are open to incorporating them.
>
> > The motivation for studying such a computation-constrained data selection problem is not fully supported.
>
> **The motivation for our study stems from a critical practical challenge in the deployment of LLMs: the computational cost of fine-tuning [Hu]**. In most practical settings, compute resources are the core constraint, and practitioners must make strategic decisions about how to allocate these resources effectively to optimize for performance.
>
> Motivated by this practical challenge, we study an important allocation problem: during fine-tuning, whether to allocate more compute to data selection or simply train on more raw data. We offer practical insights and advice based on our theoretical analysis and empirical findings.
>
> This paper encourages the community to re-evaluate the compute efficiency of existing strategies and to develop more efficient methods.
>
> *References:*
>
> [Hu] Hu, Edward J., et al. "Lora: Low-rank adaptation of large language models." *arXiv preprint arXiv:2106.09685* (2021).

---

> ### Author Response · Authors · 2024-11-21
> **Response to Reviewer mMXd (4/4): Corrections**
>
> > (Moderate) The style file seems to not follow ICLR 2025 templates: the lines are not numbered.
>
> Thank you for pointing out the formatting. We apologize for this oversight. We have since updated the manuscript to adhere strictly to the ICLR 2025 template.
>
> > (Minor) Typo(s): Figure 1 "using a much larger base model then data selection model": "then" should be "than".
>
> Thanks again for the careful read. We have fixed this in our revised version (see line 235).

---

> ### Comment · Reviewer_mMXd · 2024-11-24
>
> Thank the authors for their efforts. I'm glad to see some vital clarifications on the key concepts (which I did not fully understand before). I am not so opposed to seeing this work published as I was before, but I am still not very convinced of its significance. I will adjust my rating from 3 to 5 and leave the decision to the AC.

---

### Official Review · Reviewer_Z24H · 2024-11-04

**Soundness:** 3
**Presentation:** 3
**Contribution:** 3
**Rating:** 5
**Confidence:** 3

**Summary:**

This paper studies a framework considering the practical challenges of training and fine-tuning large language models (LLMs) under computational constraints. It has established a trade-off between achieving better performance using larger data and lowering computational costs by selecting smaller subsets of data. A key takeaway is that simpler data selection methods, such as lexicon-based and embedding-based approaches, often provide a more efficient solution compared with more complex, compute-intensive methods like perplexity-based and gradient-based strategies.

**Strengths:**

This paper addresses compute-efficient fine-tuning, which is an important task in training LLM. Extensive simulations are conducted to provide empirical evidence and support the framework.

**Weaknesses:**

1. Although the author claims some simple methods such as Lexicon outperform the complex ones such as Perplexity and Gradient, as shown in Figure 1, the complex ones perform quite well especially under medium and large budget situations. It would be more important to study the tipping point, where the performance gains plateau became flat. This is the place where further increases in computing resources yield diminishing returns.
2. It is not surprising to see the tradeoff between performance and data size. The conclusions in this paper are largely empirical and may not generalize well to other situations. The practical limit of parametric fit is limited, as it mainly fits observed data points without clear guidance on how to *extrapolate* results to new scenarios. For example, can the results from smaller models (e.g., 13B) be used to predict outcomes for larger models (e.g., 70B)? Can the parameters estimated from smaller models be reliably transferred for larger model? If practitioners need to run experiments on 70B models to obtain these insights and fit the parametric model, the results may not be useful.

**Questions:**

1. Could the author discuss a real-world scenario to demonstrate how the proposed methods could be applied to guide practitioners?
2. Are the studied methods sensitive to the choice of model architecture?
3. How do these methods scale with hardware improvements?

---

> ### Author Response · Authors · 2024-11-21
> **Response to Reviewer Z24H (1/2): The Tipping Point**
>
> Thank you for the insighful comments and careful review. Responses are divided into sections.
>
> > Although the author claims some simple methods such as Lexicon outperform the complex ones such as Perplexity and Gradient, as shown in Figure 1, the complex ones perform quite well especially under medium and large budget situations. It would be more important to study the tipping point, where the performance gains plateau became flat. This is the place where further increases in computing resources yield diminishing returns.
>
> If we unerstand the **tipping point** correctly—as the ratio between the training model size and the selection model size when complex methods like Perplexity and Gradient outperform cheaper methods like Lexicon—our empirical findings indicate that this occurs when training a 70B model using a 7B selection model. At this point, PPL and LESS outperform both BM25 and Embed across benchmarks, becoming compute-optimal for the first time. Generally, for PPL and Gradient to become compute-optimal, our extrapolation finds that the ratio between the train model size and selector model size should be greater than 5x and 10x, respectively.

---

> ### Author Response · Authors · 2024-11-21
> **Response to Reviewer Z24H (2/2): Generalization & Extrapolation**
>
> > The conclusions in this paper are largely empirical and may not generalize well to other situations.
>
> We would like to highlight the steps we've taken to ensure generalizability of our findings:
> -  We conducted extensive evaluations, over 600 training runs, across multiple model sizes—specifically 7B, 13B, and 70B parameters—and tasks, including MMLU, BBH, and IFEval, which assess various abilities of LLMs.
> -  We tested our approach on different model families, such as Llama3, and found similar trends as with the Llama2 models.
>
> Notably, many canonical works on scaling laws and compute-optimal training have also based their conclusions on empirical findings [Kaplan, Hoffmann, Muennighoff]. We provide new insights into the compute-optimality of data selection methods for LLM finetuning. Our work encourages the community to reconsider the compute efficiency of existing data selection strategies and motivates the development of new methods that are both effective and computationally efficient.
>
> >  The practical limit of parametric fit. Extrapolation from smaller model results.
>
> Thanks for pointing out the specific uses of parametric were not made clear. To clarify, we include new discussion, analysis, and figures for extrapolation (see Section 7 and Appendix G).
>
> While we cannot accurately predict the exact downstream performance, **the parametric fit obtained from smaller models allows us to predict the ratio of training model size to selector model size needed for compute-optimality.**
>
> As an example, we use the parametric fit we obtained from the 7B-7B and 13B-7B (Train Model Size/Selector Model Size) MMLU run and analyze how much bigger the train model needs to be, under gradient data selection, to surpass the current finetune Pareto frontier. As shown in (https://ibb.co/jRXx2KD), the compute-optimal ratios we derive from both 7B and 13B are 10x and 5x, respectively, suggesting that gradient method becomes compute-optimal at about 70B parameter size when using a 7B model for data selection. This aligns with our empirical observations from the 70B experiments, showing that practitioners can predict tipping points using parametric fits obtained from 7B and 13B models.
>
>
>
> *References:*
>
> [Kaplan] Kaplan, Jared, et al. "Scaling laws for neural language models." *arXiv preprint arXiv:2001.08361* (2020).
>
> [Hoffmann] Hoffmann, Jordan, et al. "An empirical analysis of compute-optimal large language model training." *Advances in Neural Information Processing Systems* 35 (2022): 30016-30030.
>
> [Muennighoff] Muennighoff, Niklas, et al. "Scaling data-constrained language models." *Advances in Neural Information Processing Systems* 36 (2023): 50358-50376.

---

> > ### Comment · Reviewer_Z24H · 2024-11-25
> >
> > Thanks for the efforts in the rebuttal. My score remains the same!

---

### Official Review · Reviewer_9BiK · 2024-11-04

**Soundness:** 4
**Presentation:** 4
**Contribution:** 4
**Rating:** 8
**Confidence:** 4

**Summary:**

Surveys and experimentally compares different data selection methods for LLM fine-tuning, and reasonably and quantitatively concludes that only rather cheap methods that choose train samples based on some cheap similarity to the validation samples are likely to be worthwhile, but depends (of course) on how much training computation you are going to run.

**Strengths:**

I found this is an important experimental contribution for practitioners and academics alike, and is likely to be heavily cited in the future. While there will inevitably be some discussion of whether they compared to all the right and best methods, I think that's in the details: they compared good and sufficiently recent example methods from high level strategies and showed significant enough differences that seem endemic to these different strategies.

**Weaknesses:**

The weaknesses I detail below should all be corrected, but they are all minor, none of them individually or in total would be a good reason to reject the paper.

SECTION 3 PROBLEMS:

At the beginning of Section 3: “Our goal is to find the optimal subset S ⊆ X” pretty sure you mean subset S ⊆ D there?

I think you are implying that the train set is not necessarily IID with the validation set, but that the validation set is IID with the test set. All I see you say is that the validation set is “correlated” with the test set, which is a really  weak and vague thing to say, but if that’s all you want to say, okay, but I will be curious if in your experiments you actually make the Val and Test sets IID.

You need to define that $T$ represents a training procedure, you just use it without defining it now.

“By ranking the data points….” Given a large initial train set D, having to rank the datapoints at cost O(D log D) is not free, hope you guys are taking that into account. Of course, you might argue just touching all D samples is O(D), but that is less relevant if, say, we have an infinite generator of data D (e.g. a real-time reader of the datastream formerly known as Twitter) and an independent (non-ranking) decider of whether each incoming $x$ is worth training on, that is, we shouldn’t have to assume we need to sort at cost O(D log D).


I’m uncomfortable as a reader that in (2) you are still defining your objective in terms of the test set. I agree that’s the ultimate goal, but if you actually implemented (2) it assumes knowledge of the test set. By the time you get to (2), I expected you to have switched to the validation set in the stated objective, which is different than the final metric, which should of course than be on the test set.


SECTION 4 FEEDBACK:
You can cut some of the intro to Sec 4, but please add-in that Lexicon-based and Embedding-based are both strategies that try to select train samples that are similar to the validation samples, whereas Perplexity and Gradient solutions are optimizing for the effect on the model loss.

SECTION 5 FEEDBACK:
Why do you assume training on all x is equal? Is that really true (honest question)? My guess is yes due to the very beaurocratic nature of how these models are run, but that’s not always true of machine-learned models, for example, a classic decision tree is much faster to evaluate for some inputs than others (if it has leaves of varying depths).

In computing C(k), you sum over C_v(x), which I assume is for x \in D? Please be explicit there about which x you are summing over.  And I’m surprised that that cost does depend on $x$ Does C_v(x) really vary so much? Could that not just be \|D\| (= size of D) times some cost per training sample?


RANDOM: Really appreciate you comparing to just a random sample as a baseline.

MINOR BUT IMPORTANT QUIBBLES:
Authors state too unequivocally: “in practice, the total compute budget is
predetermined: the number of accelerators and their usage hours are allocated in advanced”.   That certainly is NOT true in many large companies that are actively training and leading with LLMs. So please hedge and preface that sentence with “In many cases,”.


This sentence didn’t make sense to me:
“For example work on parameter efficient fine-tuning, targets improving the
memory-usage of this stage (Hu et al., 2021).”

TYPO “create an minimal” -> “a minimal”

Since you are citing John 1975, consider  also citing the famous and foundational Hart 1968 paper on Condensed Nearest Neighbors, the canonical and classic approach for selecting data for training (summarized e.g. in wikipedia: https://en.wikipedia.org/wiki/K-nearest_neighbors_algorithm). However, nearest neighbors is such a different machine learning paradigm that if you don’t feel that adds value to this work, you can ignore this suggestion, but given the statement in your paper is “Data
selection is a foundational approach in machine learning where the objective is to create an minimal
dataset from a collection of data” the Hart 1968 paper is exactly the classic paper to have done just that.

TYPO “Data selection takes the full training data as input and
chooses a subset to the train” -> “to train”

This sentence doesn’t quite parse, please re-write “Instruction-tuned models can handle a variety of possible inputs for downstream use cases as
either classification or generative model”

This sentence needs some polishing of singular vs plural:
“Therefore
while instruction-tuning is not the direct focus of this work, it provide a real-world applications of
compute-constrained data selection.”


“Assuming we at minimal” -> “at minimum”

Equation (2) can be written all on one line, winning you a bit of precious space.

In 4.1 you refer to Section 4.1, which is a bit weird and really you can just delete that whole sentence.

Do figure out how to bold the last column title C_forward in Table 1. It can be done (probably \mathbf{}).

TYPO: Fit of Compute-Performace Relationship  -> Performance

**Questions:**

The value of the strategies that depend on similarity of samples to validation samples worry me in that they seem very dependent on the size of the validation set, and that if the validation set is too small one might overfit.  But perhaps it doesn't matter too much since you are always selecting some large number of training samples anyways, and so even if the validation set is as small as Paris (to use a 2D example), you still correctly pick the subset of training samples in Europe and dump the ones in the Americas, and that wouldn't have changed much if the validation set was all of France instead of just Paris. Be great to see some discussion & experiments about this, even if they are tiny, in this paper.

See also the weaknesses section.

**Details Of Ethics Concerns:**

No ethics concerns.

---

> ### Author Response · Authors · 2024-11-21
> **Response to Reviewer 9BiK (1/4): Section 3**
>
> Thank you for a thorough, careful, and insightful review. Responses are divided into sections.
>
> > At the beginning of Section 3: “Our goal is to find the optimal subset S ⊆ X” pretty sure you mean subset S ⊆ D there?
>
> Thank you for pointing that out. We have corrected this typo (see Section 3, line 112).
>
> >  I think you are implying that the train set is not necessarily IID with the validation set, but that the validation set is IID with the test set. All I see you say is that the validation set is “correlated” with the test set, which is a really weak and vague thing to say, but if that’s all you want to say, okay, but I will be curious if in your experiments you actually make the Val and Test sets IID.
>
> Yes, we meant to say that validation set is IID with the test set, and that the train set is correlated but not necessarily IID with the validation/test set. We have rephrased the description in our assumption (see Section 3, line 122-123).
>
> > “By ranking the data points….” Given a large initial train set D, having to rank the datapoints at cost O(D log D) is not free, hope you guys are taking that into account. Of course, you might argue just touching all D samples is O(D), but that is less relevant if, say, we have an infinite generator of data D (e.g. a real-time reader of the datastream formerly known as Twitter) and an independent (non-ranking) decider of whether each incoming x is worth training on, that is, we shouldn’t have to assume we need to sort at cost O(D log D).
>
> We agree that sorting train set D cost at minimal O(D log D) after a data selection method has scored through the train set D. However, the cost to sort is minimal in comparison with the two computational bottlenecks: (1) cost of training LLMs (2) computing the utility function on D. For context, the FLOPs needed for a single forward pass on the smallest 7B model is about 4.69E+10 FLOPs per token, and finetuning for 10% of data (~about 10 Million tokens) in our settings requires 4.69E+18 FLOPs. In practice, the cost to sort poses no difference to our analysis.

---

> ### Author Response · Authors · 2024-11-21
> **Response to Reviewer 9BiK (2/4): Section 4 & 5**
>
> > I’m uncomfortable as a reader that in (2) you are still defining your objective in terms of the test set. I agree that’s the ultimate goal, but if you actually implemented (2) it assumes knowledge of the test set. By the time you get to (2), I expected you to have switched to the validation set in the stated objective, which is different than the final metric, which should of course than be on the test set.
>
> We agree with the sentiment that the objective should be defined in terms of the validation set instead of the test set. We’ve revised the objective function to align with practical implementation (see Section 4, line 145–147).
>
> > SECTION 4 FEEDBACK: You can cut some of the intro to Sec 4, but please add-in that Lexicon-based and Embedding-based are both strategies that try to select train samples that are similar to the validation samples, whereas Perplexity and Gradient solutions are optimizing for the effect on the model loss.
>
> Yes, this makes categorization much cleaner. Added additional explanation "While lexicon and embedding-based methods aim to select training samples similar to validation samples, perplexity and gradient-based methods focus on optimizing their effect on model loss. " (see line 203-205)
>
> > SECTION 5 FEEDBACK: Why do you assume training on all x is equal? Is that really true (honest question)? My guess is yes due to the very beaurocratic nature of how these models are run, but that’s not always true of machine-learned models, for example, a classic decision tree is much faster to evaluate for some inputs than others (if it has leaves of varying depths).
>
> This a good point, but it really is true in modern language models: they process inputs in batches and pad each sequence to a fixed length.

---

> ### Author Response · Authors · 2024-11-21
> **Response to Reviewer 9BiK (3/4): Miscellaneous Corrections**
>
> > MINOR BUT IMPORTANT QUIBBLES: Authors state too unequivocally: “in practice, the total compute budget is predetermined: the number of accelerators and their usage hours are allocated in advanced”. That certainly is NOT true in many large companies that are actively training and leading with LLMs. So please hedge and preface that sentence with “In many cases,”.
>
> Agree. Fixed (see line 25).
>
> >  This sentence didn’t make sense to me: “For example work on parameter efficient fine-tuning, targets improving the memory-usage of this stage (Hu et al., 2021).”
>
> Rephrased to "For example, parameter-efficient finetuning methods like LoRA (Hu et al., 2021) aim to reduce memory usage during fine-tuning by updating only a small subset of the model's parameters." (see line 34-36).
>
> > TYPO “create an minimal” -> “a minimal”
>
> Fixed (see line 38)
>
> >  Citing Hart 1968 paper "Condensed Nearest Neighbors"
>
> This is a great suggestion. Added the reference (see line 39).
>
> > TYPO “Data selection takes the full training data as input and chooses a subset to the train” -> “to train”
>
> Fixed (see line 84-85).
>
> > This sentence doesn’t quite parse, please re-write “Instruction-tuned models can handle a variety of possible inputs for downstream use cases as either classification or generative model"
>
> Rephrased to "Instruction-tuned models can handle a variety of possible inputs and can be applied to downstream tasks requiring either classification or open generation" (see line 100-101).
>
> > “Assuming we at minimal” -> “at minimum”
>
> Fixed (see line 142-143).
>
> > In 4.1 you refer to Section 4.1, which is a bit weird and really you can just delete that whole sentence.
>
> Good catch! Deleted.
>
> >  Do figure out how to bold the last column title C_forward in Table 1. It can be done (probably \mathbf{}).
>
> Fixed (see line 162-163).
>
> >  TYPO: Fit of Compute-Performace Relationship -> Performance
>
> Fixed (see line 414).

---

> ### Author Response · Authors · 2024-11-21
> **Response to Reviewer 9BiK (4/4): Questions**
>
> > The value of the strategies that depend on similarity of samples to validation samples worry me in that they seem very dependent on the size of the validation set, and that if the validation set is too small one might overfit. But perhaps it doesn't matter too much since you are always selecting some large number of training samples anyways, and so even if the validation set is as small as Paris (to use a 2D example), you still correctly pick the subset of training samples in Europe and dump the ones in the Americas, and that wouldn't have changed much if the validation set was all of France instead of just Paris. Be great to see some discussion & experiments about this, even if they are tiny, in this paper.
>
> This is indeed a valid concern. Like the reviewer has mentioned, since we are selecting large samples from the training set, the precision required is not stringent. We provide the statistics of evaluation datasets in our experiments below. This follows the hypothesis; it appears as long as the validation set is more than a few dozen (>50), BM25 and Embed should work fine.
>
> | Dataset | # Shot | # Tasks | D_val | D_test | Answer Type     |
> | ------- | ------ | ------- | ----- | ------ | --------------- |
> | MMLU    | 5      | 57      | 285   | 18,721 | Letter options  |
> | BBH     | 3      | 23      | 69    | 920    | COT and answer  |
> | IFEval  | 1      | -       | 50    | 541    | Open Generation |
>
> We've since included the evaluation dataset statistics and discussion about the size of the validation set in the revised paper (see Appendix D.4).

---

> > ### Comment · Reviewer_9BiK · 2024-11-24
> > **Reviewer acknowledgement**
> >
> > I have read the other reviews and the author response. I stand by the opinions expressed in my original review, and look forward to a spirited discussion with my colleagues about whether these studies are well-done and significant enough.

---

### Author Response · Authors · 2024-11-21
**Response to All Reviewers**

We thank each reviewer for their thorough reading of our work and their thoughtful feedback. We have made the following updates to the paper:

**Revised main paper:**

- **(Section 7)**: Added details on the fine-tuned Pareto efficient frontier and power-law fitting.
- **(Section 7)**: Added extrapolation from our fitted parametric functions to predict the compute-optimal ratio between training and selection model sizes.
    - **Findings**: For perplexity data selection, compute-optimality occurs when the training model is about 5 times larger than the selection model (35B parameters). For gradient data selection, it occurs when the training model is about 10 times larger than the selection model (70B parameters).
- Rephrased sentences throughout and fixed typos.

**Added additional empirical results:**

- **(Appendix F)**: Conducted 13B model experiments on target task IFEval, which evaluates the model's instruction following ability.
  - **Findings:** We find that the results are consistent with our previous empirical findings: at medium budget (13B model scale), cheap lexicon-based methods (BM25) and embedding-based methods (Embed) outperform perplexity-based (PPL) and gradient-based methods (LESS).

**Added appendix sections:**
- **(Appendix D.4)**: Provided details on evaluation datasets and a discussion about the size of the validation dataset.
- **(Appendix G)**: Included details, figures, and results on extrapolating from the parametric functions

**Improved appendix sections:**
- **(Appendix H)**: Added additional break-even analysis for 7B and 13B model sizes on the target task IFEval.
- **(Appendix I)**: Included additional data similarity heatmaps for the target task IFEval.

**Reviewers have asked about the practical motivations for the parametric fit**:
- **Choosing Data Selector Size.** Fits from smaller model experiments enable us to predict the train/selector model size ratio. For example, a 10x ratio derived from our 7B and 13B experiments suggests that gradient-based methods become compute-optimal at ~70B parameters when using a 7B selector model.
- **Choosing Number of Training Points.**  Fits also indicate how many data points are needed for a data selection method to achieve near-optimal performance.

---

### Meta-Review · Area_Chair_VHa3 · 2024-12-20

**Metareview:**

The paper studies the cost of data selection by formalising the trade-off between data selection cost and training gain. The authors validate their insights experimentally. They show for instance that one has to pay attention to the computational cost of sophisticated model selection techniques and that simpler (and cheaper) methods might be preferable in practice. Authors revised the paper and appendices based on the feedback of the reviewers, addressing their comments and providing additional experimental evidence.

**Additional Comments On Reviewer Discussion:**

The authors adequately clarified the concerns raised by the reviewers. After rebuttal, all reviewers trended towards acceptance. As noted by one of the reviewers during the discussion, it is not surprising to see the tradeoffs discussed in this work, but as another reviewer indicated, few works in data selection show that powerful data selection techniques are costly from a computational point of view and thus not always useful in practice. Overall, the reported results makes this paper interesting to the community and worth sharing.

---

### Decision · Program_Chairs · 2025-01-22

Accept (Poster)